# Mapping the nucleolar proteome reveals a spatiotemporal organization related to intrinsic protein disorder

Lovisa Stenström[1] , Diana Mahdessian[1], Christian Gnann[1,2], Anthony J Cesnik[2,3], Wei Ouyang[1], Manuel D Leonetti[2], Mathias Uhlén[1] , Sara Cuylen-Haering[4], Peter J Thul[1] & Emma Lundberg[1,2,3,*]

## Abstract

The nucleolus is essential for ribosome biogenesis and is involved in many other cellular functions. We performed a systematic spatiotemporal dissection of the human nucleolar proteome using confocal microscopy. In total, 1,318 nucleolar proteins were identified; 287 were localized to fibrillar components, and 157 were enriched along the nucleoplasmic border, indicating a potential fourth nucleolar subcompartment: the nucleoli rim. We found 65 nucleolar proteins (36 uncharacterized) to relocate to the chromosomal periphery during mitosis. Interestingly, we observed temporal partitioning into two recruitment phenotypes: early (prometaphase) and late (after metaphase), suggesting phase-specific functions. We further show that the expression of MKI67 is critical for this temporal partitioning. We provide the first proteome-wide analysis of intrinsic protein disorder for the human nucleolus and show that nucleolar proteins in general, and mitotic chromosome proteins in particular, have significantly higher intrinsic disorder level compared to cytosolic proteins. In summary, this study provides a comprehensive and essential resource of spatiotemporal expression data for the nucleolar proteome as part of the Human Protein Atlas.

**Keywords** human protein atlas; intrinsic protein disorder; nucleolus; perichromosomal layer

**Subject Categories** Methods & Resources; Organelles

**Mol Syst Biol. (2020) 16: e9469**

## Introduction

One of the most prominent nuclear substructures is the nucleolus, the cellular site for ribosome synthesis and assembly. In addition, the nucleoli comprise proteins involved in cell cycle regulation and stress response (Visintin & Amon, 2000; Boisvert *et al*, 2007).

Nucleoli form around nucleolar organizing regions on the ribosomal DNA sites (rDNA). As opposed to membrane-bound organelles, nucleoli and other nuclear bodies lack enclosing membranes. This allows for dynamic cellular responses as these structures can change in size and protein composition when needed. The size and number of nucleoli change throughout the cell cycle as they fuse together, a process recently suggested to be aided by interactions with the nucleoplasm (Caragine *et al*, 2019). The formation of these membrane-less yet spatially distinct structures is the result of reversible liquid–liquid phase transitions similar to oil-in-water emulsions (Brangwynne *et al*, 2009, 2011; Lin *et al*, 2015). In interphase, the nucleolus is structurally partitioned into three droplet-like layers with different miscibility (Feric *et al*, 2016). This separation facilitates a sequential production of ribosomes, from transcription of rDNA at the fibrillar center border (FC) followed by rRNA processing in the dense fibrillar component (DFC) and ribosome assembly in the granular component (GC). Phase separation is a dynamic process dependent on external factors such as pH, temperature, protein posttranslational modifications (PTMs) but most importantly protein composition and concentration. One common trait among proteins forming liquid-like droplets has shown to be the presence of low complexity sequence domains (LCDs) and protein disorder. Intrinsically disordered proteins (IDPs) are characterized by being fully or partially unfolded, making them flexible in terms of interaction, and have been suggested to be a strong driver of phase separation (Li *et al*, 2012; Berry *et al*, 2015; Elbaum-Garfinkle *et al*, 2015; Molliex *et al*, 2015; Nott *et al*, 2015). Intrinsically disordered proteins have shown to be central in various diseases such as cancer, cardiovascular diseases, and Alzheimer's disease. Mutations in disordered regions can drastically change the conformation of the protein, and since many IDPs function as hub proteins, altered protein function could initiate a loss-of-function cascade in the cell (Uversky *et al*, 2008).

The inherently high density of the nucleoli enables its isolation and purification. Several studies using mass spectrometry (MS)-based proteomics have been focused on identifying proteins residing

1 Science for Life Laboratory, School of Engineering Sciences in Chemistry, Biotechnology and Health, KTH Royal Institute of Technology, Stockholm, Sweden
2 Chan Zuckerberg Biohub, San Francisco, CA, USA
3 Department of Genetics, Stanford University, Stanford, CA, USA
4 Cell Biology and Biophysics Unit, European Molecular Biology Laboratory, Heidelberg, Germany
  *Corresponding author. Tel: +4687906000; E-mail: emma.lundberg@scilifelab.se

in the nucleoli (Andersen *et al*, 2002, 2005; Scherl *et al*, 2002; Leung *et al*, 2006). Together, they assign a nucleolar localization to over 700 proteins. Another intriguing finding is that the nucleolar proteome seems dynamic rather than static and contains many types of proteins not only related to ribosome biogenesis, indicating that the nucleolar proteome may be larger and more diverse than previously expected.

As of today, there has been no effort to spatiotemporally map the human nucleolar proteome and its subcompartments throughout the cell cycle. When the nucleolus disassembles during mitosis, most nucleolar proteins leak to the cytoplasm. However, a majority of the mitotic chromosomal mass is not chromatin but other proteins residing in the perichromosomal layer including at least 50 known nucleolar proteins (Gautier *et al*, 1992a,b,c; Angelier *et al*, 2005; Van Hooser *et al*, 2005; Takata *et al*, 2007; Ohta *et al*, 2010; Booth *et al*, 2016). One example is the proliferation marker MKI67, a highly disordered nucleolar protein shown to be important for chromosome segregation by acting as an emulsifying shield around the chromosomes in mitosis (Booth *et al*, 2014; Cuylen *et al*, 2016).

In this study, we used an antibody-based microscopy approach to generate a spatiotemporal map of the human nucleolar proteome in interphase and mitosis. We present a resource containing localization data for 1,318 nucleolar proteins including spatial sublocalization to nucleolar subcompartments such as the fibrillar center and the nucleoli rim, accessible as part of the Human Protein Atlas (HPA) database (www.proteinatlas.org; Uhlen *et al*, 2010; Thul *et al*, 2017). We also propose that the nucleoli rim is a dynamic nucleolar subcompartment with a distinct proteomic composition. Additionally, we show evidence for 65 nucleolar proteins being recruited to the chromosomal periphery during mitosis. Based on this subcellular map, we performed the first systematic analysis of intrinsic protein disorder for the human nucleolar proteome, experimentally confirming what has been conceptualized by others (Brangwynne *et al*, 2011; Nott *et al*, 2015; Feric *et al*, 2016), that a majority of the proteins have long intrinsically disordered domains.

# Results

### A detailed spatial map of the nucleolar proteome

Despite the nucleolus being an intensively studied organelle, there is currently no resource offering a complete map of the human nucleolar proteome. To address this, we used the immunofluorescence (IF) and confocal microscopy workflow developed within the HPA Cell Atlas to systematically map all human nucleolar proteins (Thul *et al*, 2017). Out of the 12,393 proteins included in the HPA Cell Atlas (v19), we identified 1,318 nucleolar proteins, of which 287 localized to the fibrillar center or dense fibrillar component (from now on collectively denoted as fibrillar center) and 1,031 localized to the whole nucleolus (Dataset EV1). A schematic image of the tripartite nucleolar organization is shown in Fig 1A, while Fig 1B and C highlight typical confocal images of proteins localizing to whole nucleoli and fibrillar centers. UTP6, a protein thought to be involved in pre-18S rRNA based on its yeast homolog, is known to localize to nucleoli (Dragon *et al*, 2002; Fig 1B). UBTF activates RNA polymerase I-mediated transcription in the fibrillar center (Kwon & Green, 1994; Fig 1C). Functional enrichment analysis of the nucleolar proteome

shows that the enriched Gene Ontology (GO) terms for biological process are well in line with the known functions of the nucleoli (e.g., ribosome biogenesis, rRNA processing, and transcription; Dataset EV2). We demonstrated the robustness of distinguishing nucleolar subcompartments by extracting features from all images in the HPA Cell Atlas using a machine-learning model (Ouyang *et al*, 2019), and then visualizing those features using a UMAP, a uniform manifold approximation and projection for dimensionality reduction (preprint: McInnes *et al*, 2018). The clustering of the IF microscopy images for the different nucleolar substructures confirms that they show distinct IF staining patterns that are consistent and robust across cell lines and antibodies, and the distance between the clusters shows that they easily can be distinguished both from each other and from other punctate nuclear substructures, such as nuclear speckles and nuclear bodies (Fig 1D).

### Most nucleolar proteins are multilocalizing

One advantage with image-based proteomics is the ability to study the *in situ* protein localization in single cells, including multilocalizing proteins (proteins localized to multiple compartments concurrently). In total, 54% of all proteins in the HPA Cell Atlas are detected in more than one cellular compartment, while as much as 87% of the nucleolar proteins ($n = 1,145$) are reported as multilocalizing ($P < 2.2 \times 10^{-16}$, using a one-tailed binomial test). Multilocalizing nucleolar proteins are visibly scattered throughout a majority of the organelle clusters in the UMAP (Fig 1D), with 33% simultaneously localizing to other nuclear locations only. However, this level of multilocalization is similar for other nuclear structures, such as the nuclear membrane and other nuclear bodies. The actin filament and plasma membrane proteome also have above 80% multilocalizing proteins (Thul *et al*, 2017). The most commonly shared localization for nucleolar proteins is the nucleoplasm, cytosol, or mitochondria.

Multilocalizing nucleolar protein observations also provide the first evidence for nucleolar localization for several proteins. LEO1 is a nucleoplasmic protein which is part of the PAF1 complex. It is involved in transcription but has no previous experimental evidence for a nucleolar localization (Rozenblatt-Rosen *et al*, 2005; Zhao *et al*, 2005). However, using both antibodies and a GFP-tagged cell line we detected LEO1 in the nucleoplasm and the fibrillar center (Fig 1E, antibody staining in wild-type cells Appendix Fig S1).

A protein association network of the shared proteomes between the nucleolus and cytosol shows a tightly connected cluster, indicating association with the same biological functions (Appendix Fig S2). Given that nucleoli synthesize and assemble ribosomes for export to the cytoplasm, multilocalizing nucleolar and cytosolic proteins could be involved in translation. The ribosomal protein cluster in the core of the network supports this. One such example is the ribosomal protein RPL13 (Fig 1F, siRNA antibody validation; Appendix Fig S3). The high number of multilocalizing nucleolar proteins suggests a functional versatility of these proteins, likely not only relating to ribosome biogenesis.

### The nucleolar proteome is larger than previously thought

Based on our data, the nucleolus comprises around 7% of the human proteome, a higher number than previously proposed

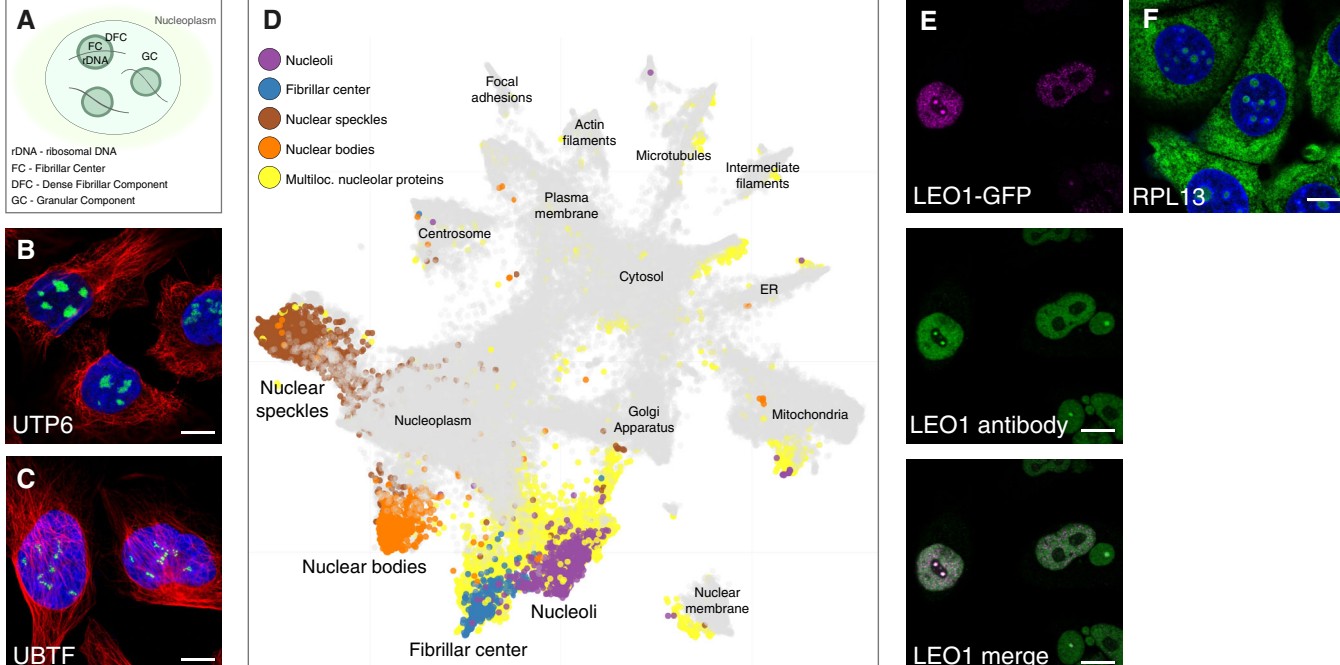

**Figure 1. A detailed spatial map of the nucleolar human proteome.**

A  Schematic overview of the nucleolus and its substructures: fibrillar center (FC), dense fibrillar component (DFC), and granular component (GC).

B  UTP6 in U-2 OS cells exemplify proteins localized to whole nucleoli (HPA025936).

C  Fibrillar center localization shown by a UBTF IF staining in U-2 OS cells (HPA006385).

D  UMAP visualization of the IF images generated in the HPA Cell Atlas (also shown in Fig 3A). The images from singularly localizing nuclear proteins are highlighted in purple (nucleoli), blue (fibrillar center), brown (nuclear speckles), and orange (nuclear bodies). Multilocalizing nucleolar proteins are highlighted in yellow.

E  Dual localization of LEO1 to both the fibrillar center and nucleoplasm in GFP-tagged HeLa cells (magenta), also supported by IF antibody staining using HPA040741 (green).

F  The multilocalizing ribosomal protein RPL13 detected in the nucleoli, cytosol, and ER in MCF-7 cells (HPA051702).

Data information: Protein of interest is shown in green, nuclear marker DAPI in blue, and the reference marker of microtubules in red. Scale bar 10 μm.

using MS-based methods (Andersen *et al*, 2002, 2005; Scherl *et al*, 2002; Leung *et al*, 2006). To assess the data reliability, we introduced a scoring system for each protein localization: enhanced, supported, approved, or uncertain (Thul *et al*, 2017). The score depends not only on available additional validation methods, such as siRNA knockdown, but also similarity in immunostaining patterns between independent antibodies or consistency with available experimental localization data in the UniProtKB/Swiss-Prot database (The UniProt Consortium, 2019). The requirements for each score are described in Materials and Methods, and the distribution of localization scores for the nucleolar proteins is shown in Appendix Fig S4.

At least 700 proteins have been reported as nucleolar in the literature (Andersen *et al*, 2002, 2005; Scherl *et al*, 2002), while currently only 266 nucleolar proteins are reported as experimentally verified in GO (Ashburner *et al*, 2000; Gene Ontology Consortium, 2019). This highlights the need for an updated resource detailing the nucleolar proteome. We compared the HPA dataset to nucleolar proteins detected in other studies (Andersen *et al*, 2002, 2005; Scherl *et al*, 2002). To simplify the comparison, these datasets were merged since they show a high overlap (Appendix Fig S5). A total of 550 proteins are reported as nucleolar in these studies but not in the HPA Cell Atlas, possibly because of obsolete gene nomenclature

resulting in incomplete mapping, potentially impure nucleolar fractions, or a lack of a specific antibody in the HPA library. In total, 237 of the HPA nucleolar proteins are verified in the studies mentioned above (Appendix Fig S5, Dataset EV1). When adding data from Orre *et al* (2019) and the experimentally verified nucleolar proteins in GO (Binns *et al*, 2009), 72 additional nucleolar proteins could be confirmed. Taken together, 1,009 nucleolar proteins are uniquely identified by the HPA dataset (Dataset EV1). Of these, 292 are not expressed in HeLa cells, which explains why they are absent from the mass spectrometry analysis on HeLa lysates (Andersen *et al*, 2002, 2005; Scherl *et al*, 2002). The HPA cell line panel consists of 64 cell lines expressing 98% ($n$ = 19,216) of all protein-coding genes, enabling the detection of most proteins. The reasons why the HPA data contains so many uniquely identified nucleolar proteins are presumably the ability to detect multilocalizing proteins with lower abundance in the nucleolus and the ability to pick up single-cell variations obscured by bulk methods. Although it cannot be ruled out that the HPA dataset also contains false-positive nucleolus localizations, these proteins should at a maximum be a subset of the category labeled with the "Uncertain" reliability score ($n$ = 122). Note that, no protein data is included in the HPA Cell Atlas, if only contradictory localization data exists.

A total of 541 of the HPA nucleolar proteins have no previous human experimental data reported for any cellular component in GO, such as the nucleolar protein KRI1 (Fig 2A and Appendix Fig S6

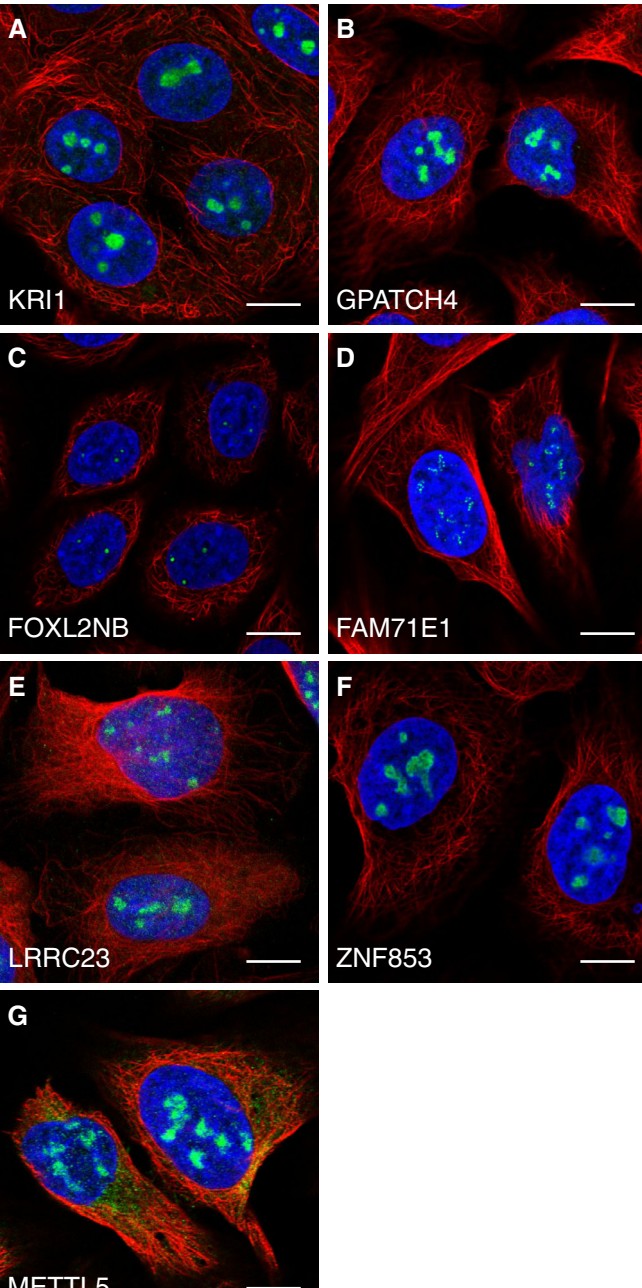

**Figure 2. Previously unknown nucleolar proteins.**

A  KRI1 localized to the nucleoli in MCF-7 cells (HPA043574).
B  GPATCH4 localized to the nucleoli in U-2 OS cells (HPA054319).
C  FOXL2NB localized to the fibrillar center in SiHa cells (HPA061017).
D  FAM71E1 localized to the fibrillar center in U-2 OS cells (HPA048111).
E  LRRC23 localized to the nucleoli in U-2 OS cells (HPA057533).
F  ZNF853 localized to the nucleoli in U-2 OS cells (HPA067690).
G  METTL5 localized to the nucleoli in U-2 OS cells (HPA038223).

Data information: Protein of interest is shown in green, nuclear marker DAPI in blue, and the reference marker of microtubules in red. Scale bar 10 μm.

for independent antibody staining and siRNA antibody validation) that may be required for ribosome biogenesis based on data from its yeast homolog (Sasaki *et al*, 2000; Huh *et al*, 2003). GPATCH4 has no localization data reported in GO, and we can validate it being nucleolar using two independent antibodies (Fig 2B and Appendix Fig S6 for independent antibody staining). The previously uncharacterized proteins FOXL2NB and FAM71E1 were localized to the fibrillar center (Fig 2C and D, Appendix Fig S6 for FOXL2NB-independent antibody staining), and LRRC23, ZNF853, and METTL5 (Fig 2E–G and Appendix Fig S6 for LRRC23-independent antibody staining) were localized to the whole nucleolus.

## The nucleolus rim has a distinct proteomic composition and can be considered a subcompartment of the nucleolus

Despite the lack of an enclosing membrane, 157 of the nucleolar proteins show a characteristic rim-like staining pattern in at least one cell line, denoted as nucleoli rim (Dataset EV1). When highlighting the single localizing nucleoli rim proteins in the UMAP, a distinct cluster adjacent to the nucleolar cluster appears, being significantly closer together than to the other nucleolar proteins ($n_{rim} = 27$ and $n_{nucleoli} = 208$, $P = 0.0294$ using nearest neighbor analysis followed by a two-tailed binomial test). This shows that the staining pattern relates to proteins in a distinct part of the nucleolus separated from the rest of the nucleolus and fibrillar centers (Fig 3A). For instance, MKI67 as well as GNL3 localize to the nucleoli rim using independent antibodies in all cell lines stained (Fig 3B and C and Appendix Fig S7). The rim localization pattern has previously been proposed to be an antibody artifact related to abundant proteins sterically hindering the antibody to penetrate the whole nucleoli; this could in theory result in a staining gradient similar to the rim localization (Sheval *et al*, 2005; Svistunova *et al*, 2012). To investigate this, we compared transcriptomics (Thul *et al*, 2017) and MS proteomic data from the U-2 OS cell line (Beck *et al*, 2011) for the rim proteins in relation to the non-rim nucleolar proteins. The analysis shows that the rim proteins on a group level are marginally more abundant compared with the non-rim nucleolar proteins ($P = 2.975 \times 10^{-5}$, $n_{rim} = 157$, and $n_{non-rim} = 1,161$ for transcriptomics and $P = 0.01992$, $n_{rim} = 96$, and $n_{non-rim} = 613$ for proteomic data, two-tailed unpaired Wilcoxon tests). However, the expression levels between the classes are still greatly overlapping (Appendix Fig S8) with both highly and lowly expressed proteins in both groups, showing that protein abundance is not the only factor driving the rim localization. To further confirm that the rim staining is not an antibody artifact, we created two cell lines expressing mNeon-Green-tagged (mNG) MKI67 and GNL3 at endogenous levels, tagged at the N-terminus and C-termini, respectively. Live cell imaging revealed a dimmer, but still visible, rim localization for MKI67 also in the tagged cell line that was further enhanced upon PFA fixation (Fig 3D and E). The rim localization was also confirmed for GNL3 in the tagged cells (Fig 3F). Since the rim localization is visible in the tagged cell lines, we conclude that proteins localizing to the nucleolar rim cannot simply be considered antibody artifacts but are instead a sign of a fourth nucleolar subcompartment with a distinct proteomic profile. The occurrence of this staining pattern seems to be complex, and

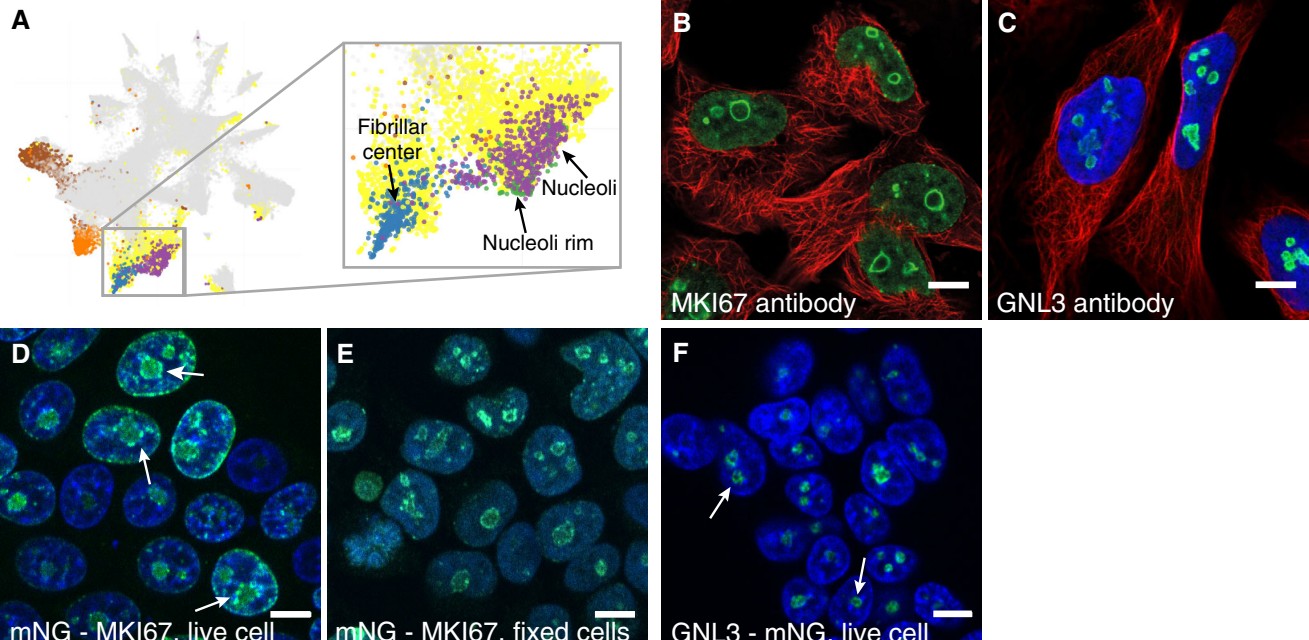

**Figure 3. The nucleoli rim localization.**

A   UMAP visualization of the IF images generated in the HPA Cell Atlas (also shown in Fig 1D), specifically highlighting the nucleolar protein clusters. The images from singularly localizing nucleolar proteins are highlighted in purple, fibrillar center proteins in blue, and nucleoli rim proteins in green. Multilocalizing nucleolar proteins are highlighted in yellow.

B   IF staining of MKI67 shows localization to nucleoli rim in U-251 cells (CAB000058).

C   IF staining of GNL3 shows localization to nucleoli rim in U-2 OS cells (HPA036743).

D   HEK 293T cells expressing endogenous levels of N-terminus mNG-MKI67 show a faint nucleoli rim localization, although still visible. White arrows indicate cells where the rim could be seen.

E   Fixed HEK 293T cells expressing mNG-tagged MKI67 show a clear nucleoli rim localization.

F   HEK 293T cells expressing endogenous levels of C-terminus tagged GNL3 show nucleoli rim localization. White arrows indicate cells where the rim could be seen.

Data information: Protein of interest is shown in green, nuclear marker DAPI/Hoechst in blue, and microtubule reference marker in red. Scale bar 10 μm.

these proteins have common molecular or functional features giving rise to this spatial organization.

## Nucleolar proteins recruited to mitotic chromosomes

As the cell enters mitosis, nucleoli are disassembled to enable the separation of the chromosomes. Most nucleolar proteins leak to the cytoplasm, while at least 50 nucleolar proteins have been shown to instead adhere to the periphery of the chromosomes (Gautier *et al*, 1992a,b,c; Angelier *et al*, 2005; Van Hooser *et al*, 2005; Takata *et al*, 2007; Ohta *et al*, 2010). To better understand the nucleolar dynamics during mitosis, we performed a single-cell spatial characterization of nucleolar proteins during cell division. MKI67 is one of the more prominent perichromosomal constituents. Thus, we generated a list of 150 targets with protein–protein association with MKI67, its interacting protein NIFK, or proteins showing a similar staining pattern in interphase as MKI67 (i.e., nucleoli rim; Dataset EV3). A mitotic shake-off protocol was used to enrich mitotic cells from an asynchronous cell population. A total of 85 nucleolar proteins could not be detected on the chromosomal periphery during cell division (Dataset EV3) as exemplified by the ribosomal protein RPS19BP1 (Appendix Fig S9). 65 proteins including MKI67 (Fig 4A) relocated to the chromosomal periphery of which 36 have, to our knowledge, no experimental data for being localized to chromosomes during cell division (Dataset EV3 and the HPA Cell Atlas, www.proteinatlas.org, for image data), exemplified by the proteins NOC2L, EMG1, BMS1, BRIX1, and RSL1D1 (Fig 4B–F, Appendix Fig S10 for independent antibody stainings of NOC2L and BMS1). Of the already known perichromosomal constituents, seven have been localized to chromosomes in chicken cells only (Ohta *et al*, 2010) and we provide experimental evidence for such a localization in human cells. For the 22 remaining known proteins, we confirmed their previously observed localization to condensed mitotic chromosomes (Gerdes *et al*, 1984; Weisenberger & Scheer, 1995; Magoulas *et al*, 1998; Westendorf *et al*, 1998; Olson *et al*, 2000; Lerch-Gaggl *et al*, 2002; Angelier *et al*, 2005; Takata *et al*, 2007; Amin *et al*, 2008; Gambe *et al*, 2009; Hirano *et al*, 2009; Hirai *et al*, 2013; Booth *et al*, 2014). Interestingly, a large fraction of the nucleoli rim proteins also relocate to the perichromosomal layer during mitosis. Assuming that the probability of a rim and non-rim protein localizing to mitotic chromosomes is equal and that the fraction of rim and non-rim proteins should be the same between the mitotic chromosome and the cytoplasmic leakage groups (73 rim proteins of 150 proteins stained, 49%), the actual distribution of rim proteins being relocated to mitotic chromosomes is significantly higher (49 of 65 mitotic chromosome proteins, 75%. $P = 1 \times 10^{-5}$, one-tailed binomial test).

## Mitotic chromosome proteins are generally more disordered than other nucleolar proteins

Intrinsic protein disorder is a known key property for the formation of membrane-less organelles, such as the nucleolus (Elbaum-Garfinkle *et al*, 2015; Molliex *et al*, 2015; Nott *et al*, 2015). We estimated the fraction of disordered residues among the nucleolar and mitotic chromosome proteins using the protein disorder prediction tool IUPred2A (Bálint *et al*, 2018). The calculated disorder fractions were used to compare the nucleolar and mitotic chromosome proteins with the cytosolic proteins as defined in the HPA Cell Atlas (v19). The nucleolar proteins show a significantly higher fraction of disordered residues per protein, with a median of 20% ($n = 848$) compared with 14% for the cytosolic proteins ($n = 2,054$, adjusted $P < 1 \times 10^{-4}$, two-tailed Kruskal–Wallis test followed by Dunn's test with Bonferroni's correction). These numbers are similar to those previously shown for the mouse proteome, where the authors concluded that the median disorder content for nucleolar proteins is 22% compared with 12% for non-nuclear proteins (Meng *et al*, 2016). This supports the hypothesis of IDP enrichment in membrane-less structures, where their ability for phase separation is needed to maintain structural integrity. Interestingly, the mitotic chromosome proteins are disordered to an even larger extent, median 36% ($n = 65$, adjusted $P < 1 \times 10^{-4}$ compared with both cytosolic proteins and with the nucleolar proteins; Fig 5A). Additionally, the nucleoli rim proteins have a slightly higher disorder level than the other nucleolar proteins with a median of 31% ($n = 153$, adjusted $P = 4 \times 10^{-4}$). The number of proteins having at least one consecutive domain of more than 30 disordered residues, a common length for functional domains, follows the same trend for the different proteomes (Fig 5B). More than 81% of the proteins localized to mitotic chromosomes have at least one disordered domain longer than 30 residues. Among the top five disordered mitotic chromosome proteins are not only the already known constituents of the perichromosomal layers CCDC86 (Ohta *et al*, 2010; known from chicken homolog data only, Fig 5C), CCDC137 (Ohta *et al*, 2010; Booth *et al*, 2014; Appendix Fig S11), and MKI67 (Gerdes *et al*, 1984; Booth *et al*, 2014; Cuylen *et al*, 2016), but also the previously unknown constituent of the perichromosomal layer, RRP15 (Fig 5D). A study shows that RRP15 is central for rRNA transcription and biogenesis. The authors further presented evidence of it being a checkpoint protein, as perturbations of RRP15 induce nucleolar stress triggering cell cycle arrest (Dong *et al*, 2017). Among the other highly disordered proteins is the uncharacterized protein PRR19, for which we show experimental data for a nucleolar localization in interphase and a relocalization to the chromosomes during mitosis (Fig 5E). We conclude that a majority of nucleolar mitotic chromosome proteins have at least one long intrinsically disordered domain and that the fraction of disordered residues among nucleolar proteins is significantly higher than for the average cellular protein.

## Half of the mitotic chromosome proteins are essential for cell proliferation

Proteins localizing to the chromosomal surface during mitosis are expected to have a function important for cellular proliferation. To

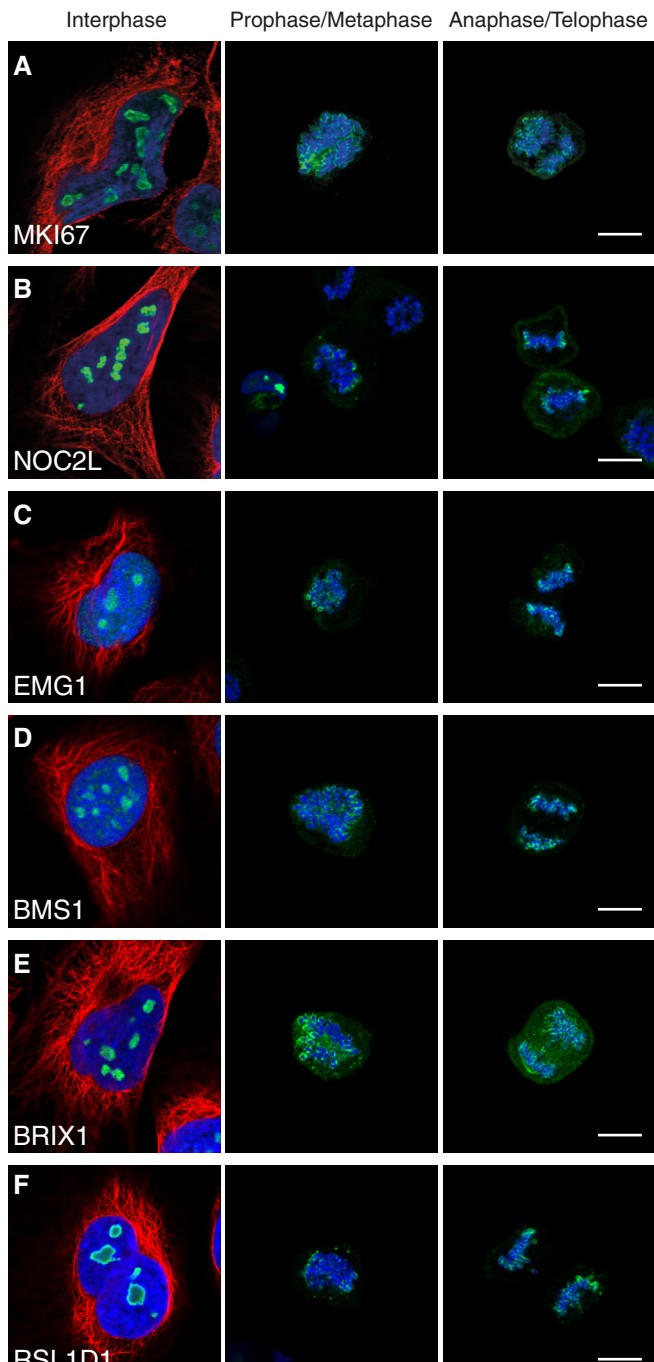

**Figure 4. Nucleolar proteins recruited to mitotic chromosomes.**

A   MKI67 (HPA000451).
B   NOC2L (HPA044258).
C   EMG1 (HPA039304).
D   BMS1 (HPA043081).
E   BRIX1 (HPA039614).
F   RSL1D1 (HPA043483).

Data information: Protein of interest is shown in green, microtubules in red, and DAPI in blue. Images of interphase cells were acquired from a different experiment, and staining intensities cannot be compared between interphase and mitotic cells. Scale bar 10 μm.

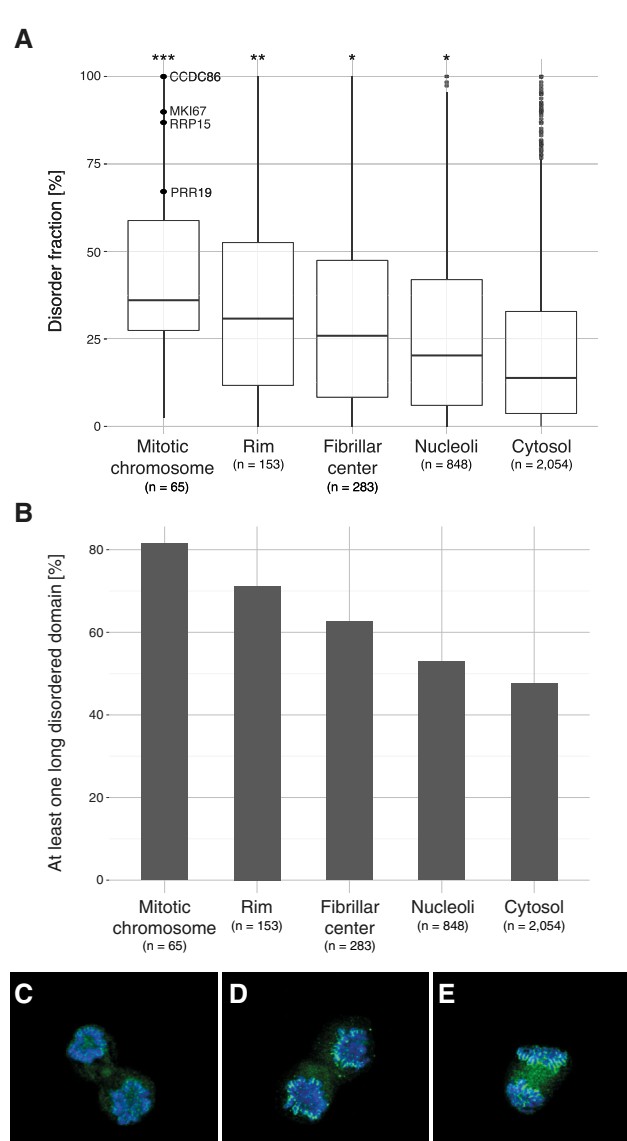

**Figure 5. Intrinsic disorder level for mitotic chromosome and nucleolar proteins as predicted by IUPRED2A.**

A   The fraction of disordered residues per protein for the nucleolar proteins is significantly higher (*, adjusted $P < 1 \times 10^{-4}$, Dunn's test with Bonferroni's correction. $n = 848$) compared with the cytosolic proteome as defined in the HPA Cell Atlas ($n = 2,054$). Additionally, the disorder levels for the nucleoli rim proteins (**, adjusted $P = 4 \times 10^{-4}$, $n = 153$) and the mitotic chromosome proteins (***, adjusted $P < 1 \times 10^{-4}$, $n = 65$) are significantly higher compared with the other nucleolar proteins. Box limits mark 1st and 3rd quantiles; whiskers, the 1.5× interquartile range; and center line, the median. Outliers are shown as dots.

B   Fraction of proteins (in percent) having at least one functional disordered domain (> 30 consecutive amino acids) for the mitotic chromosome proteins compared with the cytosolic proteome. More than 81% of the mitotic chromosome proteins have at least one long disordered domain.

C–E   IF stainings of mitotic chromosome proteins predicted to be highly disordered. Protein of interest shown in green and DAPI in blue. Scale bar 10 μm. (C) CCDC86 (HPA041117), known constituent of the perichromosomal layer, 100% predicted disorder level. (D) RRP15 (HPA024639), unknown constituent of the perichromosomal layer, 87% predicted disorder level. (E) PRR19 (HPA070350), unknown constituent of the perichromosomal layer, 67% predicted disorder level.

confirm this, we examined the essential genes across different cancer cell lines in the Achilles dataset (Meyers et al, 2017). It revealed that 34 of the mitotic chromosome genes are labeled as pan-dependent (Dataset EV3), meaning that they were ranked as the topmost depleting genes in at least 90% of the cell lines screened. This is a significantly larger fraction than for all human genes screened (2,030 of 16,383 genes, $P < 2.2 \times 10^{-16}$, Pearson's chi-squared test) and also compared to the other nucleolar genes (249 of 1,253 genes, $P = 5.3 \times 10^{-10}$, Pearson's chi-squared test). We also investigated the fraction of unfavorable marker genes predicted by the HPA Pathology Atlas, that is, genes where high expression was negatively associated with patient survival in human cancer ($P < 0.001$; Uhlen et al, 2017). The mitotic chromosome genes are significantly more prevalent ($n = 44$), both compared with all detected genes (6,835 of 19,613 genes, $P = 3.0 \times 10^{-8}$, Pearson's chi-squared test) and to the nucleolar genes not localizing to mitotic chromosomes (573 of 1,253 genes, $P = 5.4 \times 10^{-4}$, Pearson's chi-squared test). In total, 28 mitotic chromosome genes could be classified as both essential and unfavorable (Dataset EV3). For instance, BYSL, which is required for ribosome processing (Miyoshi et al, 2007), is pan-dependent, and the expression is predicted to be unfavorable in liver cancer and renal cancer. DNTTIP2 is known to regulate the transcriptional activity in the nucleolus (Koiwai et al, 2011), and we confirm its localization to human mitotic chromosomes during cell division, as previously demonstrated in chicken (Ohta et al, 2010; Appendix Fig S11). Similarly to BYSL, it is pan-dependent and the expression is unfavorable in both liver cancer and renal cancer. CCDC137 has previously been detected in the nucleolus and on the chromosomes (Ohta et al, 2010; Booth et al, 2014; Orre et al, 2019; Appendix Fig S11). Its function is unknown but is likely important during cell division, since it was both classified as pan-dependent and associated with poorer survival in renal cancer, liver cancer, and lung cancer. We conclude that half of the mitotic chromosome genes are crucial for cell proliferation of cancer cells. Some proteins, such as MKI67, are not labeled as pan-dependent, despite being known to perform essential functions during cell division. This highlights the advanced protein buffering capacity of the cell but also that half of the mitotic chromosomal proteins in our dataset are needed for maintaining cell proliferation.

## Two recruitment phenotypes among mitotic chromosome proteins revealed

Previous large-scale MS proteomic studies analyzing the proteomic composition of the perichromosomal layer were using isolated chromosomes from a bulk of cells and hence lack a temporal dimension. By using IF, we show that the recruitment of nucleolar proteins onto mitotic chromosomes can be stratified into two recruitment phenotypes, early ($n = 46$) and late ($n = 19$; Dataset EV3). MKI67 (Fig 6A) serves as an example for the early recruitment during prometaphase where an even staining intensity is maintained throughout all mitotic phases. In contrast, the recruitment of the late-onset proteins, exemplified by NOL12, TIGD1, BYSL, ACTL6B, and ZNF202, consistently occur after metaphase, suggesting phase-specific functions during mitosis (Fig 6B–F, Appendix Fig S10 for independent antibody staining of BYSL). ACTL6B has been detected on mitotic chromosomes in chicken (Ohta et al, 2010), while the other highlighted proteins are previously unknown constituents of the chromosomal periphery. In

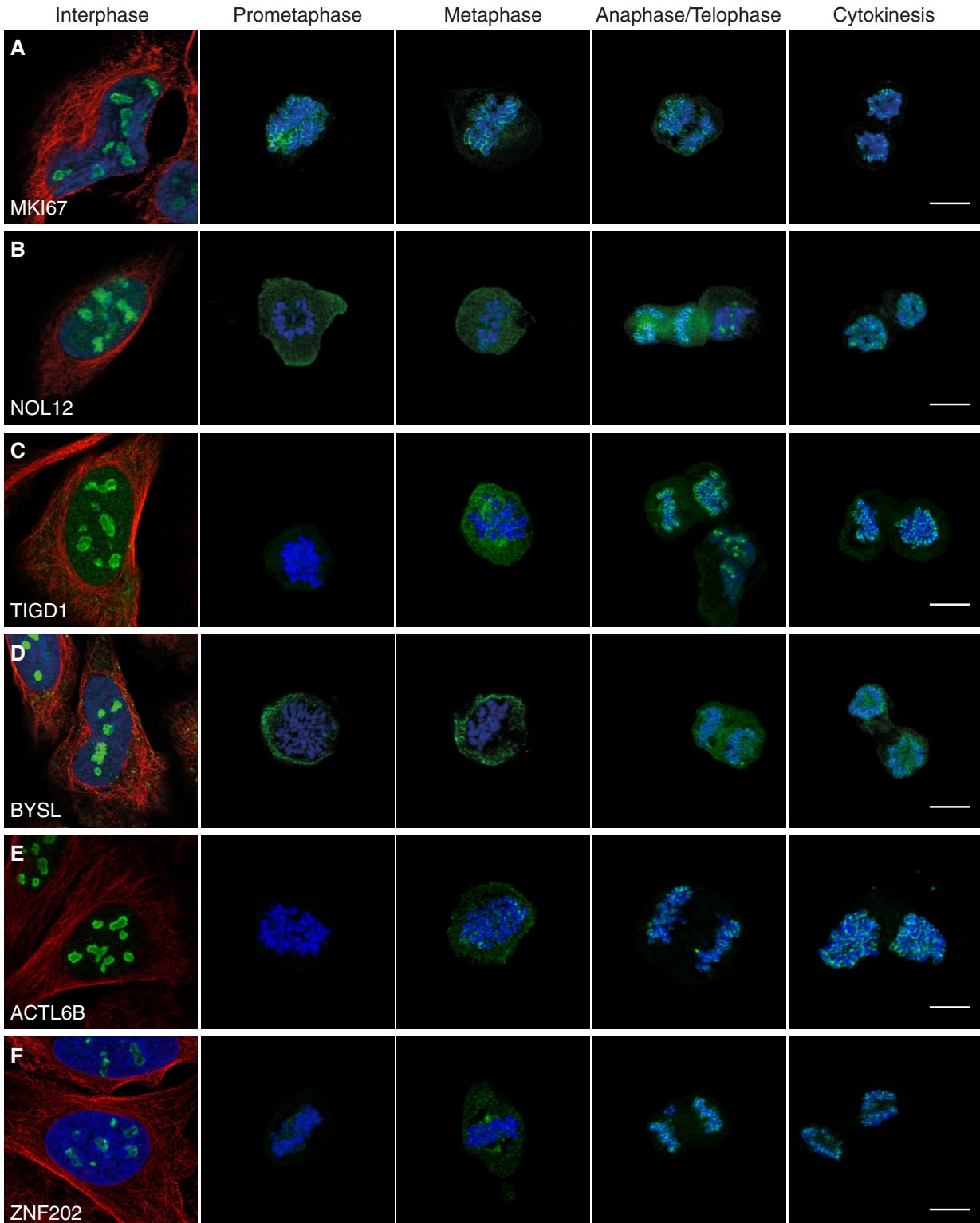

**Figure 6. Two recruitment phenotypes among mitotic chromosome proteins revealed.**

A    MKI67 is detected on the chromosomes throughout all mitotic phases and exemplifies proteins in the early recruitment category.

B–F    IF stainings of proteins where localization to mitotic chromosomes is seen after metaphase, showing a typical late recruitment expression pattern. (B) NOL12 (HPA003547). (C) TIGD1 (HPA041717). (D) BYSL (HPA031219). (E) ACTL6B (HPA045126). (F) ZNF202 (HPA059229).

Data information: Protein of interest is shown in green, microtubules in red, and DAPI in blue. Images of interphase cells were acquired from a different experiment, and staining intensities cannot be compared between interphase and mitotic cells. Scale bar 10 μm.

total, we show experimental evidence for temporal partitioning of 65 nucleolar proteins at the surface of mitotic chromosomes, with 19 proteins having a later onset in terms of their chromosomal recruitment (Appendix Fig S12).

### All mitotic chromosome proteins need MKI67 as a scaffold for binding

MKI67 is needed for proper chromosome segregation during mitosis. Previous studies have shown that the perichromosomal layer cannot form in MKI67-deficient cells, leading to a disrupted localization of the other constituents of the perichromosomal layer (Booth *et al*, 2014). That study analyzed only seven proteins, and we wanted to investigate the behavior of all 65 mitotic chromosome proteins. Therefore, we repeated the mitotic shake-off protocol using a MKI67 knockout (KO) HeLa cell line. All mitotic chromosome proteins were also stained in wild-type (WT) HeLa cells as a positive control. For 61 of the 63 proteins expressed on transcript level in WT HeLa cells, we could confirm the mitotic chromosome localization, and for 57 proteins, we could confirm the early or late recruitment (Dataset EV3 and Appendix Fig S13). As previously hypothesized by Booth *et al*, none of the proteins were fully recruited to the perichromosomal layer in the MKI67 KO cells and instead formed protein aggregates (Appendix Fig S13), as exemplified by two already known constituents of the perichromosomal layer, PES1 and GNL3 (Fig 7A and B; Lerch-Gaggl *et al*, 2002; Takata *et al*, 2007). When analyzing the level of DNA–protein colocalization in MKI67 KO and WT cells, the KO cells generally exhibited a lower overlap and a negative correlation with the DAPI staining compared with the WT cells. Significantly less overlap and/or correlation with DNA was confirmed for 48 of 63 proteins ($P < 0.05$, two-tailed unpaired Wilcoxon test, Fig 7 and Appendix Fig S13). This result shows that MKI67 depletion disrupts the recruitment of proteins to the perichromosomal layer. For 11 proteins, no mitotic chromosome localization nor protein aggregates were observed in the MKI67 KO cells, despite showing a clear mitotic chromosome location in WT cells, as exemplified by PWP1 in Fig 7C (for the remaining 10 proteins, see Appendix Fig S13 and Dataset EV3). Interestingly, the temporal partitioning with early and late recruitment to the mitotic chromosomes is less apparent in the MKI67 KO cells and can no longer be observed for 11 of the late-onset proteins (ACTL6B, BYSL, CCDC85C, CCDC86, DENND4A, DTWD1, EBNA1BP2, ETV4, MPHOSPH10, NIFK, and NOL12). For example, DENND4A and NOL12 show a disrupted mitotic chromosome localization throughout all mitotic phases (identical to the early-onset proteins PES1 and GNL3; Fig 7A and B) instead of a mitotic chromosome localization from anaphase onwards, as observed in the WT cells (Fig 7D and E). Altogether, these results suggest that the expression of MKI67 is important for the spatiotemporal control and redistribution of these proteins to the perichromosomal layer during mitosis.

## Discussion

In this study, we provide a single-cell, image-based catalogue of the nucleolar proteome with precise spatial resolution of the nucleolar substructures. This resource is made freely available as part of the HPA database (www.proteinatlas.org). We provide evidence for 541 nucleolar proteins currently lacking experimental evidence for any subcellular localization in GO (Ashburner *et al*, 2000; Gene Ontology Consortium, 2019).

We conclude that the nucleoli rim is a dynamic nucleolar subcompartment with a distinct proteomic composition. We provide evidence that it is not merely an artifact of cell fixation or antibody staining, as previously suggested (Sheval *et al*, 2005; Svistunova *et al*, 2012), because the same pattern can be observed for endogenously mNG-tagged proteins. The rim characteristic is enhanced in PFA-fixed cells, which is known to affect the solubility and charge of proteins. Rim proteins also show a tendency to be more disordered than the other nucleolar proteins, and so, these proteins might have common molecular features and functions that remain to be unraveled. We hypothesize that the nucleoli rim proteins are associated with the perinucleolar chromatin, and aid in tethering the nucleolus to the chromatin. This is supported by previous reports that MKI67 organizes heterochromatin (Sobecki *et al*, 2016). Additionally, cells depleted of NPM1, which also shows a strong nucleoli rim localization, exhibit deformed nucleoli and rearrangement of perinucleolar heterochromatin (Olausson *et al*, 2014). The authors also present data for NPM1 having a role in tethering the heterochromatin to the nucleolus and maintaining nucleolus–chromatin interactions. Another interesting finding is the overrepresentation of rim proteins being recruited to the perichromosomal layer during mitosis. A previous study shows that proteins residing in the granular component, in this case DDX21 and NCL, redistribute to the nucleoli rim upon knockdown of the nucleolar scaffold protein WDR46 (Hirai *et al*, 2013). This suggests that the localization to the nucleoli rim might be context-dependent and that further studies are needed to unravel the function of this nucleolar subcompartment.

We report a novel observation that proteins are temporally partitioned for recruitment to the mitotic chromosomes. To understand the dynamics of nucleolar temporal reorganization, we mapped a subset of the nucleolar proteins during cell division. A considerable fraction of the targeted proteins ($n = 65$) were identified on the chromosomal periphery, of which 19 showed a late recruitment to the chromosomes. This observation of temporal partitioning of protein recruitment to the mitotic chromosomes was enabled by the single-cell imaging approach and has previously been overlooked by MS proteomics. Little is known about the function of the mitotic chromosomal surface. Incorporating temporal resolution will allow further stratifications of the mitotic chromosome protein composition and reveal potential phase-specific functions. We speculate that some of the identified proteins are hitchhikers that ensure an even protein distribution in the daughter cells as the nucleolar protein aggregates in MKI67-deficient cells have shown to result in an uneven distribution of proteins after mitosis (Booth *et al*, 2014). The phenotypes of all mitotic chromosome proteins in the MKI67 KO HeLa cell line are well in line with the conclusions drawn in the previous limited study; MKI67 is required for any protein to adhere to the perichromosomal layer during mitosis.

The 19 proteins recruited to the chromosomes late during cell division could also potentially facilitate the assembly of the nucleolus during mitotic exit. Previous studies show that pre-nucleolar bodies are formed on the chromosomal periphery during telophase (Savino *et al*, 2001; Angelier *et al*, 2005; Sirri *et al*, 2016). These proteins might therefore be needed for the early reformation of the nucleolus after cell division. One essential protein for nucleoli reformation is

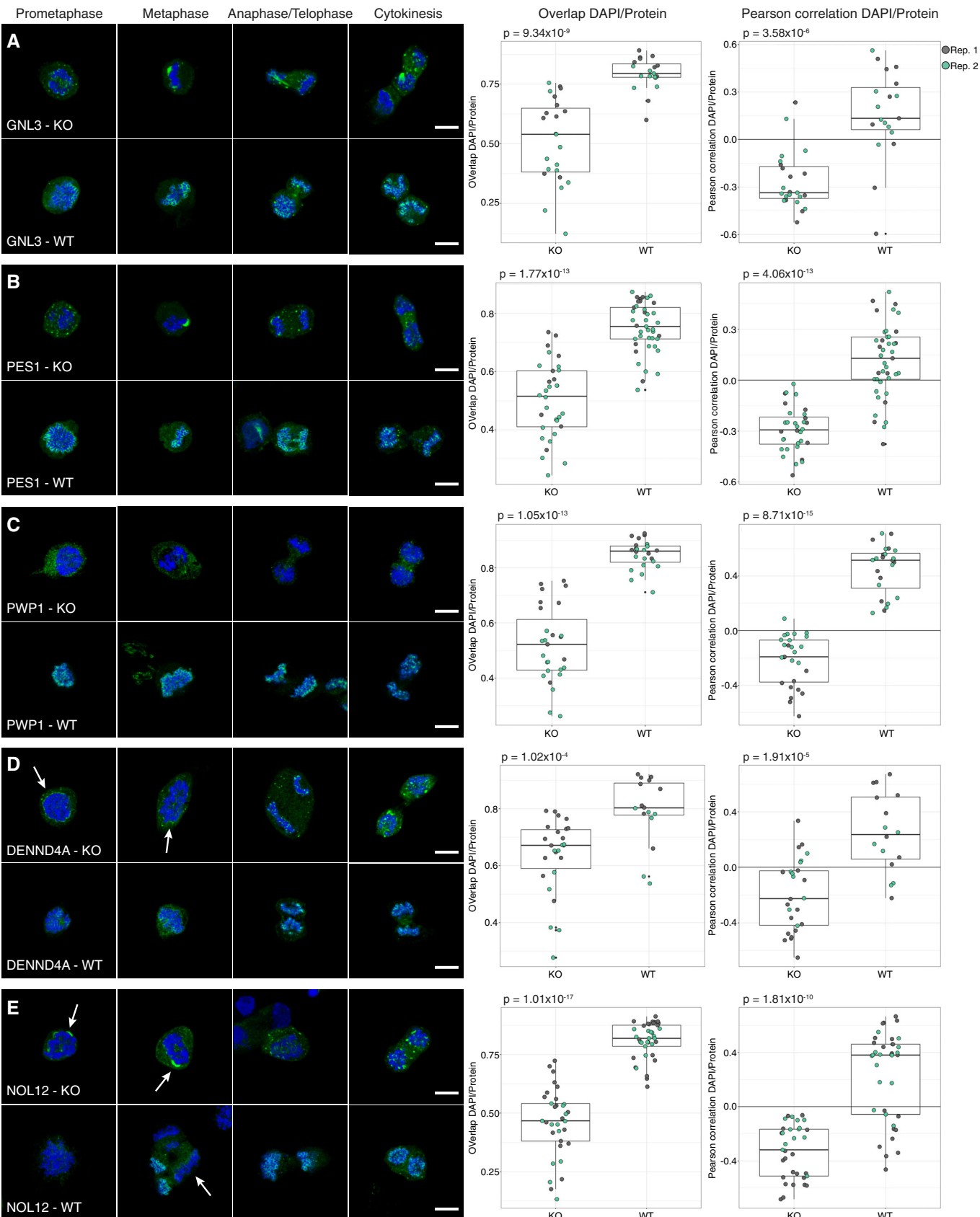

**Figure 7.**

**Figure 7. Mitotic chromosome proteins in MKI67 KO HeLa cells.**

A, B GNL3 and PES1 localize to protein aggregates around the chromosomes in the MKI67 KO cells. (A) GNL3 (HPA036743). Median overlap$_{KO}$ = 0.539, median overlap$_{WT}$ = 0.795 ($P$ = 9.34 × 10$^{-9}$), and median Pearson's correlation$_{KO}$ = −0.335, median Pearson's correlation$_{WT}$ = 0.134 ($P$ = 3.58 × 10$^{-6}$). (B) PES1 (HPA040210). Median overlap$_{KO}$ = 0.516, median overlap$_{WT}$ = 0.755 ($P$ = 1.77 × 10$^{-13}$), and median Pearson's correlation$_{KO}$ = −0.293, median Pearson's correlation$_{WT}$ = 0.129 ($P$ = 4.06 × 10$^{-13}$).

C No protein aggregates were observed in MKI67 KO cells for PWP1 (HPA038708), despite being localized to mitotic chromosomes in WT cells. Median overlap$_{KO}$ = 0.523, median overlap$_{WT}$ = 0.861 ($P$ = 1.05 × 10$^{-13}$), and median Pearson's correlation$_{KO}$ = −0.190, median Pearson's correlation$_{WT}$ = 0.515 ($P$ = 8.71 × 10$^{-15}$).

D, E Late recruited proteins do not follow the same recruitment pattern in the MKI67 KO cells compared with WT HeLa cells. White arrows indicate where protein aggregates can be seen. (D) DENND4A (HPA065343). Median overlap$_{KO}$ = 0.671, median overlap$_{WT}$ = 0.803 ($P$ = 1.02 × 10$^{-4}$), and median Pearson's correlation$_{KO}$ = −0.224, median Pearson's correlation$_{WT}$ = 0.236 ($P$ = 1.91 × 10$^{-5}$). (E) NOL12 (HPA003547). Median overlap$_{KO}$ = 0.468, median overlap$_{WT}$ = 0.820 ($P$ = 1.01 × 10$^{-17}$), and median Pearson's correlation$_{KO}$ = −0.318, median Pearson's correlation$_{WT}$ = 0.381 ($P$ = 1.81 × 10$^{-10}$). White arrow indicates metaphase cell.

Data information: Protein of interest is shown in green and DAPI in blue. Scale bar 10 μm. For the colocalization analysis data, biological replicate 1 highlighted in gray and biological replicate 2 in green. Both the overlap and Pearson's correlation between DAPI/protein staining were measured, and a two-tailed unpaired Wilcoxon test was used. Box limits mark 1$^{st}$ and 3$^{rd}$ quantiles; whiskers, the 1.5× interquartile range; and center line, the median.

EBNA1BP2 (Chatterjee *et al*, 1987), which in our data shows an increased recruitment on the mitotic chromosomes, peaking after metaphase. This suggests that proteins such as NOL12 with a similar expression pattern also are needed for nucleolar reassembly. Additionally, given our data, the late recruitment appears to be disrupted upon depletion of MKI67. If the perichromosomal layer is not able to form properly in the absence of MKI67, this could give implications in the recruitment of these proteins. Interestingly, when studying the mobility of MKI67 on the chromosomal surface using FRAP it can be seen that the mobility drops substantially during anaphase onset (Saiwaki *et al*, 2005), opening up speculations about a potential connection between the recruitment of these proteins and a stronger tethering of MKI67 to the perichromosomal layer.

We confirm the previous conceptual knowledge that human nucleolar proteins generally are more disordered than other proteins and further show that this is particularly true for those migrating to mitotic chromosomes. Our dataset is to date the most complete map of the human nucleolar proteins. As previously shown, MKI67 is highly disordered, and it is likely that other disordered proteins could play a role in coating the chromosomes and facilitating liquid–liquid interactions, given studies of the protein drivers for phase separation (Berry *et al*, 2015; Elbaum-Garfinkle *et al*, 2015; Lin *et al*, 2015; Molliex *et al*, 2015; Nott *et al*, 2015). However, it should be noted that proteins predicted to be highly disordered are not always active in phase separation as these processes are sensitive to local environment and context. Therefore, both *in vitro* and *in vivo* studies of individual proteins are needed to elucidate their capacity to promote phase separation.

Our freely available resource of the human nucleolar proteome can be used to gain better insights into the functions of the multifaceted nucleolus, such as the molecular dynamics of chromosome segregation and the role nucleolar proteins play in forming the perichromosomal layer during mitosis.

## Materials and Methods

### HPA cell atlas workflow

#### Antibody generation

Most antibodies used for the immunofluorescent experiments were rabbit polyclonal antibodies that were affinity-purified using the antigen as ligand, and validated within the Human Protein Atlas project (Uhlen *et al*, 2010). The commercial antibodies used were handled according to the supplier's recommendations. All antibodies used are listed in Dataset EV1.

#### Cell line cultivation

The cells were grown at 37°C in a 5% CO$_2$ environment. The cultivation media used for each cell line were recommended by the supplier with an addition of 10% fetal bovine serum (FBS, VWR, Radnor, PA, USA). For immunostaining, cells were seeded onto 96-well glass-bottom plates (Whatman, GE Healthcare, UK, and Greiner SensoPlate Plus, Greiner Bio-One, Germany) coated with 12.5 μg/ml fibronectin (VWR) and incubated between 4 and 20 h dependent on the cell line before fixation.

#### Sample preparation for immunofluorescent staining

An optimized protocol for proteome-wide immunofluorescent staining experiments was used, which has been described earlier in refs. Stadler *et al* (2013), Thul *et al* (2017). Each antibody was typically stained in three different cell lines, always in the bone osteosarcoma cell line U-2 OS, and in two additional cell lines having a high RNA expression of the gene (Dataset EV1 for information about cell line used for each protein). Complementary to the staining of the protein of interest, three reference markers were included: nucleus, microtubules, and endoplasmic reticulum. The cells were washed with phosphate-buffered saline, PBS (137 mM NaCl, 2.7 mM KCl, 10 mM NA$_2$HPO$_4$, 1.8 mM KH$_2$PO$_4$, pH 7.2), and then fixed by incubation with 4% paraformaldehyde (PFA, Sigma-Aldrich, Darmstadt, Germany) for 15 min. The PFA-fixed cells were then permeabilized with PBS containing 0.1% Triton X-100 (Sigma-Aldrich) for 3 × 5 min followed by another washing step with PBS. For the immunostaining, the primary rabbit mono-specific antibodies were diluted into a concentration of 2–4 μg/ml in blocking buffer (PBS + 4% FBS) containing 1 μg/ml mouse anti-alpha-tubulin (Abcam, ab7291, Cambridge, UK) and chicken anti-KDEL, respectively (Abcam, ab14234). The primary antibodies were incubated in 4°C overnight and then washed with PBS 4 × 10 min. Subsequently, blocking buffer containing 1 μg/ml of secondary antibodies (goat anti-mouse Alexa Fluor 555 (A21424), goat anti-rabbit Alexa Fluor 488 (A11034), and goat anti-chicken Alexa Fluor 647 (A-21449), all from Thermo Fisher Scientific) was added and incubated in room temperature. After 90 min, cells were counterstained with the

nuclear probe 4',6-diamidino-2-phenylindole (DAPI) diluted in PBS to 300 nM and incubated for additional 10 min. After another 4 × 10 min of washing with PBS, the glass plate was mounted with PBS containing 78% glycerol and sealed.

### Image acquisition and annotation

Image acquisition was performed using a Leica SP5 DM6000CS confocal microscope equipped with a 63× HCX PL APO 1.4 CS oil immersion objective. At least two representative images from each well were acquired using the following settings: Pinhole 1 Airy unit, 16 bit, 600 Hz, line average 2, and a pixel size of 80 nm. The gain (maximum 800 V) was kept the same for all images for one antibody. The images for each protein were then manually evaluated based on staining intensity, subcellular location, and expression pattern. The protein localization was determined manually by using the consensus for all antibodies used and cell lines stained for each protein. In total, the HPA Cell Atlas database consists of immunofluorescent stainings of more than 12,000 proteins mapped to 33 different subcellular structures. In this study, only proteins localizing to any nucleolar structure have been analyzed. Due to the image resolution, the fibrillar center annotation is used as a collective term for proteins located to either the FC or DFC of the nucleoli. Nucleoli rim and mitotic chromosome are not a public annotation in the HPA Cell Atlas (v19), but the data presented in this study will be included in HPA and can be found in Dataset EV1 and Dataset EV3, respectively.

### Classification of protein location reliability

All antibodies passing initial quality controls such as reproducibility between cell lines, correlation between signal intensity and RNA expression, reproducibility between antibodies binding different epitopes of the proteins, or agreement with validation data such as siRNA knockdown were given a reliability score. All detected localizations were given any of the following scores: enhanced, supported, approved, or uncertain. Locations within the "enhanced" category have been experimentally confirmed by at least one of the methods within the validation pillars for antibodies (Uhlen *et al*, 2016), the most common being genetic knockdown or an independent antibody. A "supported" score was given if the location in question can be confirmed by external experimental data from the Swiss-Prot database (The UniProt Consortium, 2019). If no external data is available, the location is scored as "approved", and if the location is contradictory to existing literature or that the RNA expression of the protein is low, the location got the "uncertain" label.

### Mitotic chromosome localization

To investigate nucleolar protein localization during mitosis, U-2 OS cells were cultivated in the same conditions as described above using Modified Mc Coy's 5A Medium (M8403, Sigma-Aldrich) + 10% FBS + 1% L-glutamine (G7513, Sigma-Aldrich). Before seeding onto the 96-well glass plate, the cells were gently shaken in the cultivation plate to detach the mitotic cells. After centrifugation at 500 *g* for 3 min, the mitotic cells were seeded and directly fixed with 4% PFA for 15 min at room temperature. After fixation, the IF staining protocol was carried out as described above. The 150 protein targets were chosen either by having a documented connection to MKI67 or the MKI67-interacting protein NIFK in a protein–protein association network or by showing a nucleoli rim staining similar to MKI67 in interphase. The secondary antibody used was Alexa anti-rabbit 488, and cells were also counterstained with DAPI. Images of mitotic cells in prometaphase, metaphase, anaphase/telophase, and cytokinesis were acquired with 63× magnification with the same imaging settings as previously described. Since cells were fixed, the classification of the different cell cycle phases is an estimation based on the nuclear morphology. The same gain was used to image all mitotic cells for each protein. Due to enrichment of only mitotic cells in the experiment, images of interphase cells for the same protein are taken from another experiment. Hence, the images are acquired with a different gain and staining intensities are not comparable.

## MKI67-tagged cell line generation

### Cell cultivation of Hek293T$^{mNG1-10}$ cells

Hek293T cells stably expressing mNG1-10 were cultured in high-glucose Dulbecco's modified Eagle's medium supplemented with 10% FBS, 1 mM glutamine, and 100 µg/ml penicillin/streptomycin (Gibco). Cells were maintained in 96-well plates at 37°C in a 5.0% $CO_2$ humidified environment.

### Nucleic acid reagents and gene constructs

All synthetic nucleic acid reagents (Dataset EV4) were ordered from Integrative DNA Technologies (IDT DNA). Cells were endogenously tagged at the N- or C-termini for MKi67 and GNL3, respectively. The 90-bp insert consisted of a C- or N-terminal linker, the mNG11 sequence, and a protease linker.

### RNP preparation and electroporation

Cas9/sgRNA ribonucleoprotein complexes were prepared according to established methods by assembling 100 pmol Cas9 protein and 130 pmol sgRNA (Leonetti *et al*, 2016). SgRNA was diluted in 6.5 µl Cas9 buffer (150 mM KCl, 20 mM Tris pH 7.5, 1 mM TCEP-Cl, 1 mM $MgCl_2$, 10% glycerol) and incubated at 70°C for 5 min. After addition of the Cas9 protein, the RNP assembly was performed at 37°C for 10 min. 1 µl of HDR template was added to a total volume of 10 µl. Hek293T$^{mNG1-10}$ cells were treated with 200 ng/ml nocodazole (Sigma) for 16 h prior electroporation. Electroporation was carried out according to manufacturer's instructions using SF-cell line reagents (Lonza) in an Amaxa 96-well Shuttle Nucleofector device (Lonza). Cells were washed with PBS and resuspended to $10^4$ cells/µl in SF solution. $2 × 10^5$ cells were mixed with the Cas9/sgRNA RNP complexes, electroporated using the CM-130 program, and rescued in cell culture medium in a 96 well plate. Electroporated cells were cultured and maintained for 8 days prior to FACS analysis.

### Fluorescence-activated cell sorting (FACS) and clonal enrichment

We performed analytical flow cytometry on a Sony SH800 instrument to assess the tagging efficiency of the cell lines. Singlets and live cells were analyzed. Clonal populations of the 0.5% highest mNG-expressing cells were established by sorting single cells in the wells of a 96-well plate.

### Generation of Illumina amplicon sequencing libraries

DNA repair outcomes were characterized by Illumina amplicon sequencing. Cells were grown in a 96-well plate at around 80% confluency. Medium was aspirated, and cells were washed with PBS. Cells were thoroughly resuspended in 50 µl QuickExtract (Lucigen)

and incubated at 65°C for 20 min and 98°C for 5 min. 2 μl gDNA, 20 μl 2× KAPA, 1.6 μl of 50 μM forward and reverse primers, 8 μl 5 M betaine, and 8.4 μl $H_2O$ were run using the following thermocycler settings: 3 min at 95°C followed by 3 cycles of 20 s at 98°C, 15 s at 63°C 20 s at 72°C, 3 cycles of 20 s at 98°C, 15 s at 65°C 20 s at 72°C, 3 cycles of 20 s at 98°C, 15 s at 67°C 20 s at 72°C, and 17 cycles of 20 s at 98°C, 15 s at 69°C, and 20 s at 72°C. A barcoding PCR was carried out to append unique index sequences to each amplicon for multiplexed sequencing. 1 μl of 2 nM PCR product, 4 μl of forward and reverse indexed barcoding primers, 20 μl 2× KAPA, and 11 μl $H_2O$ were run with the following thermocycler settings: 3 min at 95°C, 10 cycles of 20 s at 98°C, 15 s at 68°C, and 12 s at 72°C. Product concentrations in 200–600 bp range were quantified using a Fragment Analyzer (Agilent) and pooled at 500 nM. Amplicon sequencing library was purified using solid-phase reversible immobilization at a 1.1× bead/sample ratio. Sequencing was performed using an Illumina MiSeq system at the CZ Biohub Sequencing facility. Sequencing outcomes were characterized using CRISPResso (Pinello *et al*, 2016). Only homozygous HDR repaired cell lines or heterozygous repaired cell lines with the other allele(s) being unmodified were selected for follow-up experiments.

### Spinning disk confocal live cell microscopy

20,000 endogenously tagged HEK293T cells were grown on a fibronectin (Roche)-coated 96-well glass-bottom plate (Cellvis) for 24 h. Cells were counterstained in 0.5 μg/ml Hoechst 33342 (Thermo) for 30 min at 37°C and imaged in complete DMEM without phenol red. Live cell imaging was performed at 37°C and 5% $CO_2$ on a Dragonfly spinning disk confocal microscope (Andor) equipped with a 1.45 N/A 63× oil objective and an iXon Ultra 888 EMCCD camera (Andor).

To compare the presence of the nucleolar rim of mNG-tagged proteins in live and fixed cells, mNG-tagged cells were fixed in 4% formaldehyde (Thermo Scientific, 28908).

### MKI67 KO HeLa cell line generation

MKI67 KO HeLa cells were developed and kindly donated by Dr. Cuylen. Details about generation and validation of the cells are described in Ref. (Cuylen *et al*, 2016). WT and MKI67 KO HeLa cells were cultivated in Eagle's minimum essential medium (M4655, Sigma-Aldrich) supplemented with 10% FBS and 1% non-essential amino acids (M7145, Sigma-Aldrich) in the same conditions as mentioned above. The mitotic shake-off protocol to enrich the mitotic cells was used, and the IF and imaging protocol was followed as mentioned previously. All 65 proteins were stained in passage 3 WT HeLa and MKI67 KO cells, and 26 of these proteins were prepared in biological replicates using cells in passage 6. The same gain settings were used to image all mitotic cells for each protein, but the image intensities between the WT and the MKI67 KO HeLa cells are not comparable as they vary between the cell lines. The timing of the recruitment for BOP1, DNTTIP1, IQGAP3, and RBL2 could not be confirmed due to too few mitotic cells not covering all mitotic phases.

### Data analysis

### Gene Ontology and functional enrichment

The Web-based tool QuickGO (Binns *et al*, 2009) was used to check the overlap with the HPA nucleolar genes and nucleolar genes labeled as experimentally verified in GO (GO: 0005730, data downloaded on March 17, 2020). The GO annotations based on data from the HPA Cell Atlas were removed before comparison. To allow for the comparison of the datasets presented in Refs. (Andersen *et al*, 2002, 2005; Scherl *et al*, 2002; Orre *et al*, 2019), all detected proteins were re-mapped to current UniProt accession IDs and compared to the nucleolar proteins from the HPA Cell Atlas (v19). The functional annotation clustering for the nucleolar genes was performed using the Web-based tool DAVID (Database for Annotation, Visualization, and Integrated Discovery v.6.8; Huang *et al*, 2009a,b). All human genes were used as a background. The GO domain "biological process" terms with a Bonferroni value of < 0.01 was regarded as significantly enriched.

### Intrinsic disorder prediction

Protein disorder was predicted using IUPRED2A (Bálint *et al*, 2018). The disorder level was predicted to the nucleolar, nucleoli rim, fibrillar center, and mitotic chromosome proteins and compared to 2,054 non-nuclear cytosolic proteins as defined in the HPA Cell Atlas (v19). The mitotic chromosome proteins were removed from the nucleolar dataset prior data analysis to avoid redundancy. FASTA files were retrieved from UniProt, why a few proteins were not included in the analysis due to missing or having multiple UniProt accession IDs ($n_{rim} = 4$, $n_{FC} = 3$, $n_{nucleoli} = 10$). The output from IUPRED2A gave a probability score between 0 and 1 for each protein residue, where a score above 0.5 indicates disorder. The number of disordered residues per protein was divided by the protein length to get the disorder fraction. A Kruskal–Wallis test followed by Dunn's test with Bonferroni's correction was used to compute statistical significance.

### Functional protein association network

The functional protein association networks were created using the STRING app v 1.4.2 in Cytoscape v 3.7.1. (Doncheva *et al*, 2019).

### Nucleoli rim protein expression analysis

For the transcriptomic analysis, RNA sequencing data (in TPM) for the U-2 OS cell line generated within the HPA (available at www.proteinatlas.org) were used (Thul *et al*, 2017). For the MS proteomic analysis, the data for U-2 OS cells (measured as copies per cell) from (Beck *et al*, 2011) were used for the detected proteins, including 613 of 1,161 (53%) non-rim nucleolar proteins and 96 of 157 (61%) rim proteins. The 11 proteins annotated with "> 2e7" copies per cell were estimated to have $5 \times 10^7$ copies per cell, and the 2,109 proteins annotated with "< 5e2" copies per cell were estimated to have 100 copies per cell. A two-tailed unpaired Wilcoxon test was used to compute the statistical significance for both the transcriptomic and proteomic analyses. The UMAP was originally presented in Ref. (Ouyang *et al*, 2019) where the generation is described in detail. The nearest neighbor analysis was done using the "distances" package in R studio (v. 1.0.136). The UMAP coordinates for one image were randomly chosen to represent all images for each protein. The randomization procedure was repeated ten times, and the average number of neighbors was used in a binomial test to compare whether the number of nucleoli rim neighbors was higher than expected from the population size ($n_{rim} = 27$ and $n_{nucleoli} = 208$ for single localizing proteins).

### Pan-dependency analysis

The estimation of pan-dependent genes within the mitotic chromosome protein dataset was performed using the essential gene information from the DepMap Achilles 19Q1 Public dataset (Meyers *et al*, 2017). A pan-dependent gene is defined as "those for whom 90% of cell lines rank the gene above a given dependency cutoff. The cutoff is determined from the central minimum in a histogram of gene ranks in their 90$^{th}$ percentile least dependent line". The fraction of essential genes in the mitotic chromosome dataset and the nucleolar genes was computed, and Pearson's chi-squared test was performed to state the statistical significance.

### Prognostic cancer marker analysis

The prognostic cancer marker analysis was based on data from the HPA Pathology Atlas (Uhlen *et al*, 2017). To estimate each gene's prognostic potential, patients were classified into two expression groups based on the FPKM value of each gene and the correlation between expression level and patient survival was examined. Genes with a median expression under 1 FPKM were excluded. The prognosis of each group of patients was examined by Kaplan–Meier survival estimators, and the survival outcomes of the two groups were compared by log-rank tests. Genes with log-rank $P < 0.001$ in maximally separated Kaplan–Meier analysis were defined as prognostic genes. If the group of patients with high expression of a selected prognostic gene has a higher observed event than expected, it is an unfavorable prognostic gene; otherwise, it is a favorable prognostic gene. The full list of favorable and unfavorable prognostic genes was downloaded from the HPA website (v19, www.proteinatlas.org). The fraction of unfavorable genes within the mitotic chromosome dataset was compared to the fraction for all nucleolar genes and all human genes, respectively. Pearson's chi-squared test was carried out to compute the statistical significance.

### DNA colocalization analysis

The correlation between the HPA antibody staining and DAPI in the MKI67 KO HeLa cells compared with WT HeLa cells was measured by an image analysis pipeline in Cell Profiler v 3.1.9 (McQuin *et al*, 2018), including segmentation of the nuclear area together with the measure colocalization module. To compute the statistical significance for each measurement between WT and MKI67 KO HeLa cells, a two-tailed unpaired Wilcoxon test was used. To avoid confounding interphase or apoptotic cells present in the images, all mitotic nuclei were manually selected prior analysis. For a handful of proteins, the measured overlap and correlation pattern between WT and MKI67 KO cells was deviating. This is due to rare but unavoidable image features caused by (i) the spherical shape of the mitotic chromosomes (its whole volume does not always fit within one focal plane resulting in a partial visualization of the mitotic chromosome staining in the WT cells), and (ii) the protein aggregates in the MKI67 KO cells could end up on top of the DAPI channel, creating the illusion that the DAPI and protein staining overlap.

### Figure generation

Plots were generated using R (v. 3.6.1), R studio (v. 1.0.136), and the additional ggplot2 package. The image montages were created using FIJI ImageJ (v. 2.0.0-rc-69/1.52n).

## Data availability

- Interphase nucleolar and mitotic chromosome protein images: HPA Cell Atlas (www.proteinatlas.org).
- Nucleolar protein dataset: Dataset EV1.
- Mitotic chromosome dataset: Dataset EV3.

**Expanded View** for this article is available online.

## Acknowledgements

We acknowledge the entire staff of the Human Protein Atlas program. We also thank staff at the Chan-Zuckerberg Biohub and the Allen Institute for Cell Science for fruitful discussions on the nucleolus. Funding was provided by the Knut and Alice Wallenberg Foundation (2016.0204) and Swedish Research Council (2017-05327) to E.L.

## Author contributions

EL conceived the study. EL and LS developed methodology and carried out data investigation. LS conducted experimental work and data analysis. DM advised and conducted experimental work for the mitotic shake-off protocol and data analysis. CG and MDL generated and imaged tagged cell lines for nucleoli rim investigation. AJC performed analysis of mass spectrometry data, and WO performed machine-learning-based image analysis. SC-H developed and provided the MKI67 KO and WT HeLa cell lines and advised experimental work and data analysis. LS created the figures. EL and LS did manuscript writing. AJC, CG, DM, MU, PJT, SC-H, and WO revised the manuscript. EL supervised and administered project and funding.

## Conflict of interest

The authors declare that they have no conflict of interest.

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
