## [Review Process File · Molecular Systems Biology]

Mapping the nucleolar proteome reveals a spatiotemporal organization related to intrinsic disorder

Emma Lundberg, Diana Mahdessian, Mathias Uhlen, Lovisa Stenström, Christian Gnann, Peter Thul, Manuel Leonetti, Sara Cuylen-Håring, Anthony Cesnik, and Wei Ouyang

DOI: [10.15252/msb.20209469](https://doi.org/10.15252/msb.20209469)

Corresponding author(s): Emma Lundberg (emma.lundberg@scilifelab.se)

Review Timeline:

Submission Date:	23rd Jan 20
Editorial Decision:	3rd Mar 20
Revision Received:	24th May 20
Editorial Decision:	30th Jun 20
Revision Received:	2nd Jul 20
Accepted:	3rd Jul 20

Editor: Maria Polychronidou

Transaction Report:

Thank you again for submitting your work to Molecular Systems Biology. We have now heard back from the two referees who agreed to evaluate your study. As you will see below, the reviewers raise substantial concerns, which unfortunately preclude the publication of the study in its current form.

Both reviewers point out that while the study presents a potentially relevant resource, as it stands it remains rather preliminary and in absence of substantial follow up analyses and experimentation several of the main conclusions are not well supported. As such, they rated the conclusiveness as "Low" and were not enthusiastic about publication of the study in MSB. During our cross-commenting process (in which the reviewers are given the opportunity to comment on each other's reports), reviewer #2 mentioned that they agree with reviewer #1's concerns, namely "that the assertions of the paper are over-stated, that the data presented would benefit from additional quantification, statistical analysis, and transparent sharing of reagents used in supplementary data" and they thought that "reviewer #1's point about more carefully comparing microscopy-identified nucleolar proteins vs. proteomics-identified and database-annotated nucleolar proteins is valid". During the cross-commenting process, we specifically consulted with the reviewers asking whether they think that the study should be given a chance for revision, given that it is clear that at this point it is rather preliminary. Both reviewers reiterated that in their opinion rather substantial revisions would be required, but that if you were willing to perform such extensive revisions, they would not be opposed to evaluating a new version of the work. Taken together, we have decided to offer you a chance to address the issues raised in a major revision.

Without repeating all the points listed below, the more fundamental issues that need to be convincingly addressed are the following:

- More comprehensive comparisons of the identified nucleolar proteins with other datasets need to be included.
- Further support needs to be provided for the proposed role of MKI67 in the recruitment of nucleolar proteins on mitotic chromosomes.
- Additional experimental evidence is required to support the conclusion that the perichromosomal layer is liquid-like. Such experiments would significantly strengthen the impact of the study and we would strongly encourage you to include them. Reviewer #3 provides constructive suggestions in this regard.
- The novelty of the claim that "nucleolar proteins are found to be more disordered" needs to be toned down, and the claim needs to be contextualized with previous reports of similar findings.

All issues raised by the reviewers need to be satisfactorily addressed. As you may already know, our editorial policy allows in principle a single round of major revision, so it is essential to provide responses to the reviewers' comments that are as complete as possible. I understand that the required revisions are substantive. Please feel free to contact me in case you would like to discuss in further detail any of the issues raised or if you would like to share your revision plan with me.

On a more editorial level, we would ask you to address the following issues.

Reviewer #1:

Summary

The manuscript by Stenström presents a detailed analysis of the nucleolar proteome based on analysis of confocal images data obtained from the Human Protein Atlas. This analysis allowed the nucleolar localization of 1318 human proteins. For a subset of these proteins (287), a more detailed distribution within the nucleolus was defined by distinguishing proteins that localize to fibrillar components and the nucleoplasmic border of the nucleolus. Further, the authors analysed the partitioning of nucleolar proteins during different phases of mitosis and suggest that MKI67 is a regulator of this partitioning using a HeLa MKI67 knock-out cell line. Finally, the authors speculate on the biophysical properties of the perichromosomal layer by analyzing the content of disordered regions in nucleolar vs. cytosolic proteins.

General remarks

The topic covered by the manuscript, nucleolar proteome and its dynamics, is of general interest. However, the analysis provided in the manuscript is insufficient to support the claims made by the authors. This work expands on previous data based on mass spectrometry analysis of isolated nucleoli (correctly cited by the authors in the introduction) to compile a more comprehensive resource of the nucleolar proteome. In addition, for a subset of proteins (287), it achieves sub-organelle resolution by distinguishing between proteins associated to the fibrillar component of the nucleolus vs. proteins associated with the nucleoplasmic border. However, given the presented data, it is rather difficult to appreciate the advancement that this work provides relatively to state of the art. Similarly, regarding the role of MKI67 in organizing the recruitment of mitotic chromosome proteins, the authors extend previous analysis (done on 7 proteins) to a larger number of mitotic chromosome proteins (65) and claim that the recruitment of the majority of them was disrupted by MKI67 knock out. However, the authors present only representative pictures for a subset of these proteins showing often a single cell. Although promising, the analysis appears too preliminary making it impossible to judge whether the data support the conclusion made by the authors. Finally, the statement that the perichromosomal layer is liquid-like is supported exclusively by the observation that nucleolar proteins tend to have a higher disorder level compared to cytosolic proteins.

Major points

1. The authors observed a poor overlap between their dataset and the Gene Ontology annotation provided by UniProt (which apparently comprise only 264 nucleolar proteins) (Supplementary Figure 5). However, this analysis is very limited. For example, already using the annotation "Subcellular location [cc]" in UniProt provides 407 reviewed entries annotated with "nucleolus" localization. The authors should do a much more comprehensive comparisons of the overlap between their dataset and different types of annotations as well as a direct comparison against the 700 proteins that they state have been previously identified by proteomics. The analysis should indicate whether there is a significant overlap between dataset as well as a description of the proteins that do not overlap. It would be of general interest to understand why some proteins have been classified as nucleolar using imaging-based approaches, but have been missed using proteomics of isolated nucleoli. Related to this, how about proteins defined as nucleolar using other approaches but not by imaging analysis? Can the authors comment on these proteins and ideally estimate the rate of false negative in their analysis due to, e.g., the antibody not being available or cell type specific protein expression? Apparently, most of the work was done in U-2 OS.

2. The authors state that several nucleolar proteins are localizing to other cellular compartments, however they do not provide any comparison to proteins from other organelles. In this way, it is impossible for the reader to judge whether multilocalization is a distinctive feature of nucleolar proteins or not. For example, by checking "The multilocalizing proteome" in the Human Protein Atlas webpage <https://www.proteinatlas.org/humanproteome/cell/multilocalizing> it appears that also nuclear membrane proteins have a similar proportion of multilocalizing proteins (85%). Therefore, the authors should put this analysis into context and avoid over-statement.

3. Regarding the role of MKI67 in the recruitment of nucleolar proteins to mitotic chromosome, (Figure 7), the authors should provide for each of the 65 proteins analysed a proper a statistical analysis of protein localization performed on a sufficiently large number of cells, ideally from independent biological replicates (cells from different passages).

4. The claim that this study shows for the first time that nucleolar proteins are more disordered compared to other proteins (Page 16, Discussion) is not novel, as indicated by the authors themselves in the result section (Meng et al. 2016, Page 11). This needs to be corrected to do not mislead the readers.

5. The authors should provide additional experimental evidence to claim that the perichromosomal layer is liquid-like, or this statement should be removed.

Minor points

- The authors should indicate in Supplementary Table S1 which antibody was used for each individual proteins and whether the antibody is commercially available.
- The authors should more clearly specify which cell lines were used for their analyses. In the Method section they state: "Each antibody was typically stained in three different cell lines, always in the bone osteosarcoma cell line U-2 OS and in two additional cell lines having a high RNA expression of the gene". What does typically mean? The authors should report in supplementary table s1 on which cell lines each gene was tested. How the information from different cell lines were combined?
- Supplementary Figure 2 is unreadable, please correct.
- Supplementary Figure 8. The authors compare the abundance of nucleolar rim-localized and other nucleolar proteins using gene expression data. They should use available estimates of protein abundance for U2-OS (see for example: <https://www.ncbi.nlm.nih.gov/pubmed/22068332>), since they state themselves that "there is no perfect correlation between mRNA and protein level in cells". They should also test if there is a significant difference between the two groups.
- Page 14, the sentence "As opposed to in the WT cells...." does not make sense and contains several typos, e.g., "...mostly kept in it's liquid-phase...". Please re-phrase.

Reviewer #2:

Summary

This study uses an automated immunostaining and microscopy workflow to identify proteins that localize to the nucleolus. The authors determine that a substantial portion of the cellular proteome (~7%) can be found in the nucleolus. There is cell-to-cell and cell cycle variability in this localization, suggesting that the composition of the nucleolus is even more complex and dynamic than was known based on previous proteomic studies. The authors identify ~150 proteins that localize to the

edge of the nucleolus in immunostained cells. As this type of localization has previously been ascribed to fixation and/or antibody accessibility artifacts, the authors show with 2 example proteins that endogenously tagged nucleolar proteins show a similar localization to the edge of the nucleolus. The authors also profile the localization of nucleolar proteins through the cell cycle and determine that >60 nucleolar proteins relocate to the surface of the chromosomes during mitosis. This type of localization has been previously reported for the abundant nucleolar protein KI67, and the authors show some evidence that the association of other nucleolar proteins to the surface of mitotic chromosomes depends on the presence of KI67. This localization and dependence on KI67 has been reported in the past for four nucleolar proteins (Booth et al., 2014) but the authors expand this analysis to a much larger number of proteins. To begin to understand the biophysical features of nucleolar proteins that enable their multi-localization within cells, the authors analyze the extent of intrinsic disorder in nucleolar proteins and determine that nucleolar proteins are significantly more disordered as a class than cytosolic proteins are, consistent with similar observations made for the mouse proteome.

This study makes rigorous use of an automated microscopy approach to profile the behaviors of the nucleolar proteome. This dataset will be a helpful resource for biologists studying ribosome biogenesis, nucleolar biology, and nuclear cell biology. The strength of the study is in its breadth and the sheer number of proteins profiled; intriguing models are posed but are not definitively explored, which limits the impact of the study. Findings that are broader but conceptually similar to previous reports (KI67-dependent localization of nucleolar proteins to the mitotic chromosome periphery; intrinsic disorder of nucleolar proteins) also limit the impact of the study.

Major Points

1. A previous study, which the authors cite, reports that multi-localizing and nucleolar proteins are more likely to be intrinsic disordered (Meng et al 2016). In light of this, it is an overstatement to say in the discussion that "We show for the first time on a proteome-wide level that nucleolar proteins in general, and those migrating to mitotic chromosomes in particular, are more disordered compared to other proteins".
2. The authors state that KI67 is required for localization of other nucleolar proteins to the mitotic chromosome mass - including both an early-localizing and late-localizing group of proteins. While example images are shown, this assertion would be strengthened by quantifying the extent of DNA colocalization for these proteins over stages of mitosis in wild type vs. KI67-null cells.
3. Based on the biophysical features of KI67 and other nucleolar proteins that localize to the mitotic chromosome mass, the authors propose that this interface is "liquid-like" in nature as the nucleolus is. While an intriguing possibility, the authors do not directly demonstrate this. It is possible that KI67 and nucleolar proteins behave quite distinctly in mitosis vs. in interphase, as they are extensively phosphorylated. There are also examples of gel-sol transitions of intrinsically disordered proteins in different contexts (for instance, the protein FUS- see Patel, Hyman & colleagues, Cell 2015). To support the assertion that the perichromosomal layer is liquid-like, the authors could perform fluorescence recovery after photobleaching on mitotic chromosomes using the fluorescently tagged cell lines the authors have made. I have rated the validity of conclusions drawn 'low' because of the over-interpretation in this area.

In order for this manuscript to be suitable for publication, in my opinion these major points would need to be addressed by moderating the conclusions made or doing some FRAP experiments as suggested in comment #3 above. Modifying the text is an acceptable solution in my mind as the

breadth of the data in the study is impressive.

Reviewer #1:**Summary**

The manuscript by Stenström presents a detailed analysis of the nucleolar proteome based on analysis of confocal images data obtained from the Human Protein Atlas. This analysis allowed the nucleolar localization of 1318 human proteins. For a subset of these proteins (287), a more detailed distribution within the nucleolus was defined by distinguishing proteins that localize to fibrillar components and the nucleoplasmic border of the nucleolus. Further, the authors analysed the partitioning of nucleolar proteins during different phases of mitosis and suggest that MKI67 is a regulator of this partitioning using a HeLa MKI67 knock-out cell line. Finally, the authors speculate on the biophysical properties of the perichromosomal layer by analyzing the content of disordered regions in nucleolar vs. cytosolic proteins.

General remarks

The topic covered by the manuscript, nucleolar proteome and its dynamics, is of general interest. However, the analysis provided in the manuscript is insufficient to support the claims made by the authors. This work expands on previous data based on mass spectrometry analysis of isolated nucleoli (correctly cited by the authors in the introduction) to compile a more comprehensive resource of the nucleolar proteome. In addition, for a subset of proteins (287), it achieves sub-organelle resolution by distinguishing between proteins associated to the fibrillar component of the nucleolus vs. proteins associated with the nucleoplasmic border. However, given the presented data, it is rather difficult to appreciate the advancement that this work provides relatively to state of the art. Similarly, regarding the role of MKI67 in organizing the recruitment of mitotic chromosome proteins, the authors extend previous analysis (done on 7 proteins) to a larger number of mitotic chromosome proteins (65) and claim that the recruitment of the majority of them was disrupted by MKI67 knock out. However, the authors present only representative pictures for a subset of these proteins showing often a single cell. Although promising, the analysis appears too preliminary making it impossible to judge whether the data support the conclusion made by the authors. Finally, the statement that the perichromosomal layer is liquid-like is supported exclusively by the observation that nucleolar proteins tend to have a higher disorder level compared to cytosolic proteins.

We thank the reviewer for the highly constructive comments and present a manuscript revised and improved accordingly.

Major points

1. The authors observed a poor overlap between their dataset and the Gene Ontology annotation provided by UniProt (which apparently comprise only 264 nucleolar proteins) (Supplementary Figure 5). However, this analysis is very limited. For example, already using the annotation "Subcellular location [cc]" in UniProt provides 407 reviewed entries annotated with "nucleolus" localization. The authors should do a much more comprehensive comparisons of the overlap between their dataset and different types of annotations as well as a direct comparison against the 700 proteins that they state have been previously identified by proteomics. The analysis should indicate whether there is a significant overlap between

dataset as well as a description of the proteins that do not overlap. It would be of general interest to understand why some proteins have been classified as nucleolar using imaging-based approaches, but have been missed using proteomics of isolated nucleoli. Related to this, how about proteins defined as nucleolar using other approaches but not by imaging analysis? Can the authors comment on these proteins and ideally estimate the rate of false negative in their analysis due to, e.g., the antibody not being available or cell type specific protein expression? Apparently, most of the work was done in U-2 OS.

We thank that reviewer for this comment, which we wholeheartedly agree with. Unfortunately, this is not as straightforward as one would have wanted. Below we provide an in-depth examination of the overlap to other datasets.

General comments:

- The number of nucleolar proteins having a reviewed entry that is based on any assertion method in UniProt is 407. The number of reviewed nucleolar proteins with experimental evidence is 276. Since UniProt is importing data from the HPA Cell Atlas without labelling the source, we can not make use of a comparison to UniProt without risking circular arguments and inflation of our results. Therefore, we used experimentally verified nucleolar proteins in Gene Ontology (GO) instead to be able to filter out the proteins that are reported from HPA (since GO discloses their source). Up to date (March 17th 2020) GO contains 266 protein experimentally verified nucleolar proteins excluding the ones reported from HPA.
- The cited papers from isolated nucleoli are from 2002 and 2005 respectively, and the data that was included in NOPdb is no longer available. Instead, we included a comparison with the three key papers: Andersen *et al.* 2002, Andersen *et al.* 2005 and Scherl *et al.* 2002 respectively. Many of the entries reported in these studies are obsolete in UniProt today, but all of the currently existing proteins in UniProt have been included. We also compared to a more recent study by Orre *et al.* 2019 which should reflect state of the art in the MS field better.
- To simplify the comparison, the data from Andersen *et al.* 2002, Andersen *et al.* 2005 and Scherl *et al.* 2002 was merged (from now on denoted as 'A-A-S') since they show a quite high overlap (see Appendix Figure S5). All three studies were done on fractionated HeLa cells during the same time period from which you would expect at high overlap. In total the three datasets consist of 787 proteins.
- The overlap between the HPA and A-A-S and can be seen in the Appendix Figure S5 from which it shows that HPA have 1081 unique proteins not reported as nucleolar in the other three studies. The data from A-A-S presents in total 550 proteins not detected as nucleolar in HPA.

Comments on the 1081 HPA unique proteins compared to A-A-S:

- The experimentally verified nucleolar proteins in GO confirms additional 26 proteins not detected in the A-A-S datasets, leaving the number of HPA unique proteins to 1055.

- 39 proteins are predicted to be nucleolar in GO, e.g., N4BP1, EIF3L and TSEN15.
- 292 of 1055 (28%) genes are not expressed in HeLa cells which would explain why they have not been detected in the HeLa fractionations used in the other studies. The cell line panel available in HPA consists of 64 different cell lines (<https://www.proteinatlas.org/humanproteome/cell/cell+line>) that together express 98% (n = 19,216) of all protein-coding genes giving us a good opportunity to detect all nucleolar proteins. All antibodies are stained in U-2 OS but also in two other cell lines chosen based on the highest mRNA expression. Which cell lines that were used for each nucleolar protein are specified in Dataset EV1.
- 587 of 1055 proteins (56%) have no experimental data for a subcellular location at all in GO and cannot be considered as contradicting previous knowledge.
- 468 of 1055 proteins (44%) have experimental data in GO annotated for another localization than nucleoli, of those 55 % have other localizations reported together with the nucleolar in HPA that is in line with existing literature. It is known that most MS based fractionation studies struggle with multilocalizing proteins, and it is likely that these proteins are lacking from previous MS based studies for this very reason. 177 proteins have evidence for nucleus which dependent on source also could include the nucleoli. Many of the proteins contradicting previous data reported in GO have limited literature based on specific studies, where a potential nucleolar localization might not even be evaluated or validated.
- 252 of 1055 proteins were localized to the fibrillar center (88% of all fibrillar center proteins are hence unique to our dataset). It is likely that the fibrillar center is harder to detect using fractionation protocols.
- Among the 1055 unique proteins, 72 have validated antibodies according to any of the "validation pillars" (Uhlén *et al.*, 2016) referred to in the article, e.g., CCDC137 (<https://www.proteinatlas.org/ENSG00000185298-CCDC137/cell>), DHP6 (<https://www.proteinatlas.org/ENSG00000134146-DHP6/cell>) and LEO1 (<https://www.proteinatlas.org/ENSG00000166477-LEO1/cell>).
- Orre *et al.* confirms additional 46 nucleolar proteins among the 1055 HPA unique. For example ZNF202 and ZNRD1 was detected in the nucleolar fraction in Orre *et al.* but has not been identified in the other studies.

Comments on the 550 detected in A-A-S but not in HPA:

- 79 proteins are only named as "Novel nucleolar protein" 1, 2, 3... from Andersen *et al.* 2002, which might appear under another name in the HPA dataset. This leaves 471 unique proteins.
- In the study done by Andersen *et al.* 2005 reagents as actinomycin B were used to investigate the flux of proteins in the nucleoli upon change of metabolic state. The nucleolar localization of these proteins is therefore not always seen in untreated cells.
- 74 of 471 (16%) are not released in HPA v19. For 9 of them we have an antibody towards nucleoli but was not released due to information about known nucleolar

- localization was not included in GO (will be updated accordingly in next release of the HPA Cell Atlas). For 65 proteins we do not have antibodies of sufficient quality.
- 280 of the 397 proteins (70%) that are available in HPA v19 are reported to have a location within the nucleus but not specifically the nucleoli. Proteins showing a homogenous staining throughout the whole nucleus are not annotated as nucleolar but nuclear which might explain the discrepancy. Additionally, one cannot rule out the possibility that some of the proteins detected as nucleolar in the A-A-S studies are contaminants from the nucleoplasmic fraction, e.g., SF3B2 which is a known nuclear speckle protein where we have data from an siRNA validated antibody. Or PDS5A which is a known nuclear protein and we have data from two independent antibodies where one of them also have been validated with siRNA. An advantage with antibody-based approaches is the ability to easily distinguish these similar nuclear structures as seen in the UMAP added to Figure 1 where the image features for nucleolar and nuclear speckle stainings form two distinct clusters.
 - 299 of the 397 (75%) proteins available in HPA have supportive/enhanced gene reliability score, which means that we have other locations reported for the protein that is supported by other existing experimental literature or that we have used validated antibodies.

In summary, the comparison to existing data is difficult due to old datasets, multilocalizing proteins, databases no longer maintained or properly updated, usage of different cell lines etc. Due to word count limitations, we have not been able to include all the details above, but still included a significant amount in the following updated sections:

- Results pp. 7-8:
- Materials and methods pp. 23-24:
- Updated Figure 2
- Updated Appendix Figure S5

2. The authors state that several nucleolar proteins are localizing to other cellular compartments, however they do not provide any comparison to proteins from other organelles. In this way, it is impossible for the reader to judge whether multilocalization is a distinctive feature of nucleolar proteins or not. For example, by checking "The multilocalizing proteome" in the Human Protein Atlas webpage <https://www.proteinatlas.org/humanproteome/cell/multilocalizing> it appears that also nuclear membrane proteins have a similar proportion of multilocalizing proteins (85%). Therefore, the authors should put this analysis into context and avoid over-statement.

This is a valid point and we now address under the "Most nucleolar proteins are multilocalizing" headline in the results section (p. 6).

"One advantage with image-based proteomics is the ability to study the in situ protein localization in single cells, including multilocalizing proteins (proteins localized to multiple compartments concurrently). In total, 54% of all proteins in

the HPA Cell Atlas are detected in more than one cellular compartment, while as much as 87% of the nucleolar proteins (n = 1,145) are reported as multilocalizing (P < 2.2x10⁻¹⁶, using a one-tailed Binomial test). Multilocalizing nucleolar proteins are visibly scattered throughout a majority of the organelle clusters in the UMAP (Fig 1D), with 33% concurrently localizing to other nuclear locations only. However, this level of multilocalization is similar to other nuclear structures, such as the nuclear membrane and other nuclear bodies. The actin filament and plasma membrane proteome also have above 80% multilocalizing proteins (Thul et al, 2017). The most commonly shared localization for nucleolar proteins is the nucleoplasm, cytosol or mitochondria.”

3. Regarding the role of MKI67 in the recruitment of nucleolar proteins to mitotic chromosome, (Figure 7), the authors should provide for each of the 65 proteins analysed a proper a statistical analysis of protein localization performed on a sufficiently large number of cells, ideally from independent biological replicates (cells from different passages).

We have now addressed this by adding image data for all the proteins stained in WT and MKI67 KO HeLa cells as well as a DNA colocalization analysis and statistical significance for each measurement, using a two-tailed unpaired Wilcoxon test (Figure 7, Appendix Figure 13 and a text summary of the phenotype seen in Dataset EV3). We also updated the Results section (pp. 14-15) and Material and Methods (p. 26) accordingly. Data from biological replicates are shown for 26 proteins. For a handful of proteins, the measured overlap and correlation pattern between WT and MKI67 KO cells was deviating. This is due to rare but unavoidable image features caused by i) the spherical shape of the mitotic chromosomes. Its whole volume does not always fit within one focal plane resulting in an partial visualization of the mitotic chromosome staining in the WT cells and ii) the protein aggregates in the MKI67 KO cells could end up on top of the DAPI channel, creating the illusion that the DAPI and protein staining overlap.

4. The claim that this study shows for the first time that nucleolar proteins are more disordered compared to other proteins (Page 16, Discussion) is not novel, as indicated by the authors themselves in the result section (Meng et al. 2016, Page 11). This needs to be corrected to do not mislead the readers.

We have clarified that this is not novel on a conceptual level. What is novel is that we can demonstrate that this concept holds true for the nucleolar proteome based on the most complete analysis of nucleolar proteins in human cells. We have clarified in the text that we have experimentally confirmed in human cells what has been conceptualized before. The statement has now been changed in the

- Introduction (p. 4):

“Based on this subcellular map we also performed the first systematic analysis of intrinsic protein disorder for the human nucleolar proteome, experimentally

confirm what has been conceptualized by others, that a majority of the proteins have long intrinsically disordered domains.”

- Discussion (p. 17):

“We confirm the previous conceptual knowledge that human nucleolar proteins generally are more disordered compared to other proteins and further show that this is particularly evident for those migrating to mitotic chromosomes. “

5. The authors should provide additional experimental evidence to claim that the perichromosomal layer is liquid-like, or this statement should be removed.

The statement has been removed. Due to the current lab closure situation because of Covid-19 we have not been able to perform the FRAP analysis. With that said, the mobility of MKI67 during mitosis has already been tested, for instance by (Saiwaki *et al.* 2005). Interestingly the authors show that MKI67 is highly mobile on the chromosomal surface until anaphase onset, where the mobility gradually drops until interphase. This is certainly interesting since the proteins in the late onset category are recruited to the mitotic chromosomes during the same time period. However, in the lack of proof for the perichromosomal layer being liquid-like, we have instead removed the speculation regarding this matter.

Minor points

- The authors should indicate in Supplementary Table S1 which antibody was used for each individual protein and whether the antibody is commercially available.

The antibodies used have been added to Dataset EV1. The antibody supplier and antibody validation details are available at each protein page at www.proteinatlas.org.

- The authors should more clearly specify which cell lines were used for their analyses. In the Method section they state: "Each antibody was typically stained in three different cell lines, always in the bone osteosarcoma cell line U-2 OS and in two additional cell lines having a high RNA expression of the gene". What does typically mean? The authors should report in supplementary table s1 on which cell lines each gene was tested. How the information from different cell lines were combined?

The cell lines stained for each protein have been added as a separate tab in Dataset EV1. The information from the different cell lines are combined manually based on the general consensus staining (mentioned now in the methods section under the headline "Image acquisition and annotation", p. 19).

- Supplementary Figure 2 is unreadable, please correct.

A higher resolution version of Appendix Figure S2 has been added.

- Supplementary Figure 8. The authors compare the abundance of nucleolar rim-localized and other nucleolar proteins using gene expression data. They should use available estimates of protein abundance for U2-OS (see for example: <https://www.ncbi.nlm.nih.gov/pubmed/22068332>), since they state themselves that "there is no perfect correlation between mRNA and protein level in cells". They should also test if there is a significant difference between the two groups.

This is a valid point that we addressed by comparing the protein levels reported in the study suggested by the reviewer, and the result has been added under:

- Results (p. 9):
- Materials and Methods (pp. 24-25):
- Appendix Figure S8

- Page 14, the sentence "As opposed to in the WT cells...." does not make sense and contains several typos, e.g., "...mostly kept in it's liquid-phase...". Please re-phrase.

Thanks for pointing this out, the sentence has been rephrased.

Reviewer #2:

Summary

This study uses an automated immunostaining and microscopy workflow to identify proteins that localize to the nucleolus. The authors determine that a substantial portion of the cellular proteome (~7%) can be found in the nucleolus. There is cell-to-cell and cell cycle variability in this localization, suggesting that the composition of the nucleolus is even more complex and dynamic than was known based on previous proteomic studies. The authors identify ~150 proteins that localize to the edge of the nucleolus in immunostained cells. As this type of localization has previously been ascribed to fixation and/or antibody accessibility artifacts, the authors show with 2 example proteins that endogenously tagged nucleolar proteins show a similar localization to the edge of the nucleolus. The authors also profile the localization of nucleolar proteins through the cell cycle and determine that >60 nucleolar proteins relocate to the surface of the chromosomes during mitosis. This type of localization has been previously reported for the abundant nucleolar protein KI67, and the authors show some evidence that the association of other nucleolar proteins to the surface of mitotic chromosomes depends on the presence of KI67. This localization and dependence on KI67 has been reported in the past for four nucleolar proteins (Booth et al., 2014) but the authors expand this analysis to a much larger number of proteins. To begin to understand the biophysical features of nucleolar proteins that enable their multi-localization within cells, the authors analyze the extent of intrinsic disorder in nucleolar proteins and determine that nucleolar proteins are significantly more disordered as a class than cytosolic proteins are, consistent with similar observations made for the mouse proteome.

This study makes rigorous use of an automated microscopy approach to profile the behaviors of the nucleolar proteome. This dataset will be a helpful resource for biologists studying ribosome biogenesis, nucleolar biology, and nuclear cell biology. The strength of the study is in its breadth and the sheer number of proteins profiled; intriguing models are posed but are not definitively explored, which limits the impact of the study. Findings that are broader but conceptually similar to previous reports (KI67-dependent localization of nucleolar proteins to the mitotic chromosome periphery; intrinsic disorder of nucleolar proteins) also limit the impact of the study.

We thank the reviewer for the highly constructive comments and present a manuscript revised and improved accordingly.

Major Points

1. A previous study, which the authors cite, reports that multi-localizing and nucleolar proteins are more likely to be intrinsic disordered (Meng et al 2016). In light of this, it is an overstatement to say in the discussion that "We show for the first time on a proteome-wide

level that nucleolar proteins in general, and those migrating to mitotic chromosomes in particular, are more disordered compared to other proteins".

We have clarified that this is not novel on a conceptual level. What is novel is that we can demonstrate that this concept holds true for the nucleolar proteome based on the most complete analysis of nucleolar proteins in human cells. We have clarified in the text that we have experimentally confirmed in human cells what has been conceptualized before. The statement has now been changed in the

- Introduction (p. 4):
"Based on this subcellular map we also performed the first systematic analysis of intrinsic protein disorder for the human nucleolar proteome, experimentally confirm what has been conceptualized by others, that a majority of the proteins have long intrinsically disordered domains."
- Discussion (p. 17):
"We confirm the previous conceptual knowledge that human nucleolar proteins generally are more disordered compared to other proteins and further show that this is particularly evident for those migrating to mitotic chromosomes. "

2. The authors state that KI67 is required for localization of other nucleolar proteins to the mitotic chromosome mass - including both an early-localizing and late-localizing group of proteins. While example images are shown, this assertion would be strengthened by quantifying the extent of DNA colocalization for these proteins over stages of mitosis in wild type vs. KI67-null cells.

We have now addressed this by adding image data for all the proteins stained in WT and MKI67 KO HeLa cells as well as a DNA colocalization analysis and statistical significance for each measurement, using a two-tailed unpaired Wilcoxon test (Figure 7, Appendix Figure 13 and a text summary of the phenotype seen in Dataset EV3). We also updated the Results section (pp. 14-15) and Material and Methods (p. 26) accordingly. Data from biological replicates are shown for 26 proteins. For a handful of proteins, the measured overlap and correlation pattern between WT and MKI67 KO cells was deviating. This is due to rare but unavoidable image features caused by i) the spherical shape of the mitotic chromosomes. Its whole volume does not always fit within one focal plane resulting in an partial visualisation of the mitotic chromosome staining in the WT cells and ii) the protein aggregates in the MKI67 KO cells could end up on top of the DAPI channel, creating the illusion that the DAPI and protein staining overlap.

3. Based on the biophysical features of KI67 and other nucleolar proteins that localize to the mitotic chromosome mass, the authors propose that this interface is "liquid-like" in nature as the nucleolus is. While an intriguing possibility, the authors do not directly demonstrate this. It is possible that KI67 and nucleolar proteins behave quite distinctly in mitosis vs. in interphase, as they are extensively phosphorylated. There are also examples of gel-sol transitions of intrinsically disordered proteins in different contexts (for instance, the protein FUS- see Patel,

Hyman & colleagues, Cell 2015). To support the assertion that the perichromosomal layer is liquid-like, the authors could perform fluorescence recovery after photobleaching on mitotic chromosomes using the fluorescently tagged cell lines the authors have made. I have rated the validity of conclusions drawn 'low' because of the over-interpretation in this area.

In order for this manuscript to be suitable for publication, in my opinion these major points would need to be addressed by moderating the conclusions made or doing some FRAP experiments as suggested in comment #3 above. Modifying the text is an acceptable solution in my mind as the breadth of the data in the study is impressive.

The statement has been removed. Due to the current lab closure situation because of Covid-19 we have not been able to perform the FRAP analysis. With that said, the mobility of MKI67 during mitosis has already been tested, for instance by (Saiwaki *et al.* 2005). Interestingly the authors show that MKI67 is highly mobile on the chromosomal surface until anaphase onset, where the mobility gradually drops until interphase. This is certainly interesting since the proteins in the late onset category are recruited to the mitotic chromosomes during the same time period. However, in the lack of proof for the perichromosomal layer being liquid-like, we have instead removed the speculation regarding this matter.

Thank you for sending us your revised manuscript. We have now heard back from the two reviewers who were asked to evaluate the revised study. As you will see below, both reviewers think that the study has largely improved after the performed revisions and they are supportive of publication. Reviewer #1 lists a couple of remaining minor issues, which we would ask you to address in a minor revision.

We would ask you to address the following editorial issues.

REFEREE REPORTS

Reviewer #1:

I am satisfied with the greatly improved revised manuscript.

I have three small remaining comments to be addressed prior to publication:

- Page 4: "experimentally confirming what has been conceptualized by others". Please include also here citation(s) to previous work that conceptualized the relationship between intrinsic disorder and nucleolar proteome.
- Figure 1A is not very clear. Does it represent a nucleus or a nucleolus only? Also, please include a legend explaining the abbreviations.
- In addition to Appendix Figure S5, it would be useful to report the overlap between this resource and previous proteomics data in a tabular form as appendix, to make it easier for the reader to browse through which proteins overlap, or not, between the dataset.

Reviewer #2:

The authors have done a commendable job of responding to critiques especially in the current circumstances. The additional quantification in Figure 7 and in the supplement drive home the point of a broad dependence of nucleolar proteins on MKI67 for mitotic chromosome recruitment. The thorough comparison of this HPA dataset to previous proteomic datasets will be very helpful for the field. This manuscript is ready for acceptance in my opinion.

Reviewer #1 comments:

3. Page 4: "experimentally confirming what has been conceptualized by others". Please include also here citation(s) to previous work that conceptualized the relationship between intrinsic disorder and nucleolar proteome.

Thanks for noticing that the references were missing, we have added them now.

4. Figure 1A is not very clear. Does it represent a nucleus or a nucleolus only? Also, please include a legend explaining the abbreviations.

We have clarified this Figure and abbreviations.

5. In addition to Appendix Figure S5, it would be useful to report the overlap between this resource and previous proteomics data in a tabular form as appendix, to make it easier for the reader to browse through which proteins overlap, or not, between the dataset.

We have added this to the dataset EV1 where all nucleolar proteins are listed.

Thank you again for sending us your revised manuscript. We are now satisfied with the modifications made and I am pleased to inform you that your paper has been accepted for publication.

Corresponding Author Name: Emma Lundberg

Manuscript Number: MSB-20-9469